# DREAMCATALYST: FAST AND HIGH-QUALITY 3D EDITING VIA CONTROLLING EDITABILITY AND IDENTITY PRESERVATION

**Jiwook Kim**[*], **Seonho Lee**[*], **Jaeyo Shin, Jiho Choi & Hyunjung Shim**
Graduate School of Artificial Intelligence, KAIST, Republic of Korea
{tom919,glanceyes,jaeyo_shin,jihochoi,kateshim}@kaist.ac.kr

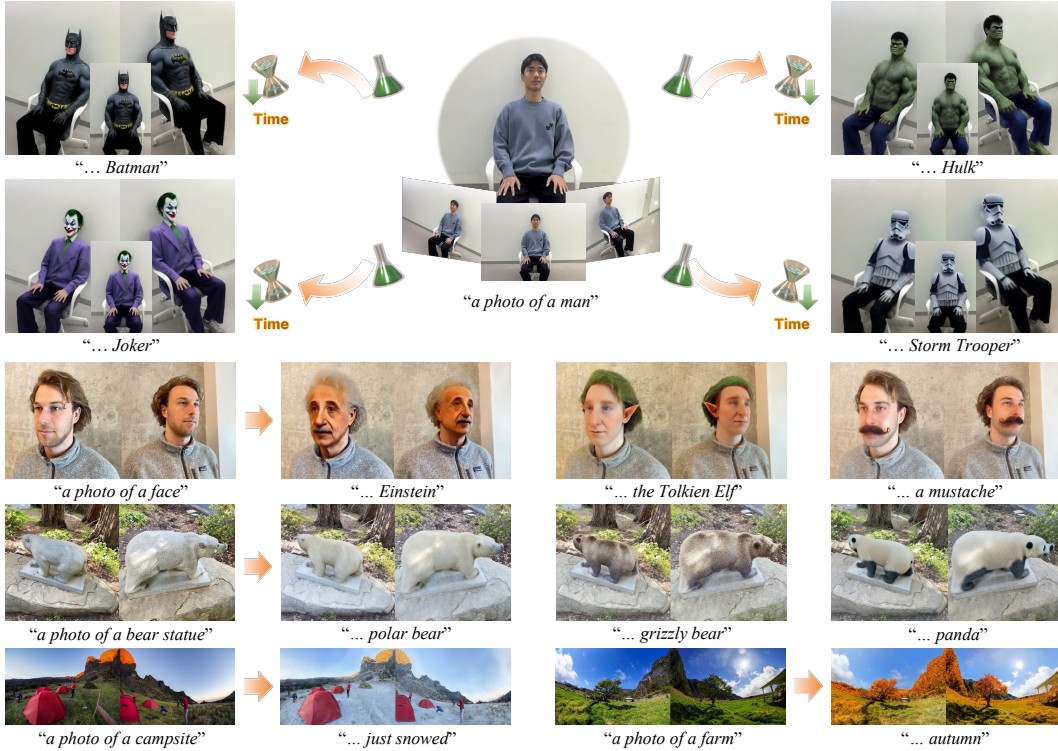

Figure 1: **Examples of 3D editing obtained by DreamCatalyst.** DreamCatalyst edits 3D scenes based on the given text prompt. DreamCatalyst not only aligns with the prompt in high-quality but also effectively preserves the identity of scenes, achieving these edits at a faster rate.

## ABSTRACT

Score distillation sampling (SDS) has emerged as an effective framework in text-driven 3D editing tasks, leveraging diffusion models for 3D-consistent editing. However, existing SDS-based 3D editing methods suffer from long training times and produce low-quality results. We identify that the root cause of this performance degradation is *their conflict with the sampling dynamics of diffusion models*. Addressing this conflict allows us to treat SDS as a diffusion reverse process for 3D editing via sampling from data space. In contrast, existing methods naively distill the score function using diffusion models. From these insights, we propose DreamCatalyst, a novel framework that considers these sampling dynamics in the SDS framework. Specifically, we devise the optimization process of our DreamCatalyst to approximate the diffusion reverse process in editing tasks, thereby aligning with diffusion sampling dynamics. As a result, DreamCatalyst successfully reduces training time and improves editing quality. Our method offers two modes: (1) a fast mode that edits Neural Radiance Fields (NeRF) scenes

[*]Equal contribution

See more extensive results on our project page: https://dream-catalyst.github.io.

approximately 23 times faster than current state-of-the-art NeRF editing methods, and (2) a high-quality mode that produces superior results about 8 times faster than these methods. Notably, our high-quality mode outperforms current state-of-the-art NeRF editing methods in terms of both speed and quality. DreamCatalyst also surpasses the state-of-the-art 3D Gaussian Splatting (3DGS) editing methods, establishing itself as an effective and model-agnostic 3D editing solution.

# 1 INTRODUCTION

Neural Radiance Field (NeRF) (Mildenhall et al., 2021) and 3D Gaussian Splatting (3DGS) (Kerbl et al., 2023) are widely used in recent text-driven 3D generation and editing. These tasks face challenges in data collection due to the need for images from diverse views of a 3D scene. Poole et al. (2022) address this issue by leveraging the rich priors of a large web-scale pretrained diffusion model (Rombach et al., 2022), proposing Score Distillation Sampling (SDS). It enables training parameterized models, especially NeRFs and 3DGS, without additional data collection.

While 3D scene generation has garnered substantial interest (Zhu et al., 2023; Tang et al., 2023; Wang et al., 2024), comparatively fewer studies have focused on 3D scene editing. The text-driven 3D editing task modifies a source scene to align with a target text prompt. Unlike 3D generation, 3D editing must consider not only alignment with the target text prompt but also identity preservation of the source scene. Posterior Distillation Sampling (PDS) (Koo et al., 2023) achieves this by considering text-aligned editability and identity preservation through minimizing the proposed stochastic latent matching loss (Wu & De la Torre, 2023; Huberman et al., 2024).

However, we observe that PDS suffers from slow 3D editing and inferior editing quality due to its theoretical foundation in stochastic latent matching. First, the formulation of stochastic latent matching heavily prioritizes identity preservation over editability at low noise perturbation as Fig. 2a. This results in insufficient editing outcomes since imperceptible details are primarily generated (Choi et al., 2022) at low noise perturbation. Second, the stochastic latent matching loss conflicts with the diffusion reverse process for editing. In recent SDS-based 3D generation studies (Zhu et al., 2023; Huang et al., 2023; Lee et al., 2024), SDS is regarded as a reverse process of diffusion models by utilizing decreasing timestep sampling to imitate the sampling procedure. This enables fast and high-quality 3D generation by sampling 3D contents with diffusion models. This new perspective allows us to design 3D generation via sampling rather than merely learning from score functions. However, directly imposing an approximated reverse process on PDS makes identity preservation challenging. In the early stages of editing, large noise perturbations hinder identity preservation of the source scene (Meng et al., 2021), making it challenging to balance between identity preservation and editability in editing tasks. For these reasons, the conflict between the diffusion reverse process and the stochastic latent matching loss leads to slower and inferior editing (see Appendix C.1). To avoid this conflict, PDS inevitably employs random timestep sampling, which differs from the diffusion reverse process. In short, editability is hampered at small timesteps, while the approximated diffusion sampling process at large timesteps reduces identity preservation, as demonstrated in Fig. 2a. The divergence of the identity preservation coefficient at low noise perturbation leads to the conflict with editability. Additionally, the information loss of source features at large timesteps further conflicts with the diffusion reverse process. Therefore, the stochastic latent matching loss has several drawbacks in balancing between identity preservation and editability.

Even if a well-designed balancing mechanism is available, there are limitations to quality improvement with reweighting formulation alone. It is because identity preservation and editability are trade-offs (Koo et al., 2023). Modifying the model architecture is a conventional solution to overcome this issue (Cao et al., 2023). Especially in SDS-based methods, many studies (Koo et al., 2023; Wang et al., 2024) finetune diffusion models to overcome the trade-offs. Specifically, several attempts have been made to improve quality with Low-Rank Adaptation (LoRA) (Hu et al., 2021) or Dreambooth (Ruiz et al., 2023). However, LoRA and Dreambooth require additional computation and training for the network. Since it leads to longer training times and additional memory costs, they are not suitable for achieving our main goal.

To address these issues, we propose (1) a novel objective function to rebalance the weights between identity preservation and editability with respect to the level of noise perturbation. Additionally, we present (2) an improved model architecture for high-quality results. For rebalancing, we in-

troduce a general formulation of PDS by introducing a new perspective of Delta Denoising Score (DDS) (Hertz et al., 2023), which is implicitly incorporated in PDS for editability. Our interpretation indicates that the objective of SDS-based editing is equivalent to the single-step of denoising and renoising in the inversion-based reverse process of SDEdit (Meng et al., 2021). Moreover, we suggest two conditions for designing the objective function to rebalance the weights. By integrating these conditions, we present a specialized formulation that considers the diffusion timestep roles and approximated diffusion reverse process to boost editing speed and improve quality.

Unlike the stochastic residual loss of PDS, our loss function provides more weight to identity preservation when timesteps are high. Meanwhile, it reduces the emphasis on identity preservation when timesteps are low. Accordingly, our loss function reweights identity preservation and editability by considering the role of diffusion timesteps for the editing task. This strategy provides two advantages. First, our loss ensures superior editing performance as fine details are synthesized. Second, the proposed loss reduces the conflict between SDS-based editing and approximated diffusion reverse process (Huang et al., 2023). This results in enhanced speed and editing quality. When the approximated diffusion reverse process is adopted to traditional methods (Hertz et al., 2023; Koo et al., 2023), significant degradation of source information occurs due to strong perturbation at the early stages of training. However, our loss function mitigates the degradation of source information by reweighting, as demonstrated in Fig. 2b.

Toward an improved model architecture, we introduce leveraging FreeU (Si et al., 2023) in SDS. It allows us to overcome the extensive computation and memory costs of LoRA and Dreambooth. FreeU does not require any time consumption and additional memory while improving quality. Moreover, FreeU enhances the editability by suppressing the high-frequency features, while preserving the identity by amplifying the low-frequency features. These characteristics of FreeU harmonize with our formulation. It is because our formulation ensures identity preservation while FreeU enhances editability without compromising identity preservation, as shown in Fig. 1.

We evaluate our method through qualitative and quantitative comparisons and user studies with baseline methods. Our results demonstrate that the proposed method outperforms the baselines in both editing speed and quality. Moreover, our method shows outstanding results on both NeRF and 3DGS. These results indicate DreamCatalyst is a model-agnostic 3D editing framework, rather than a method specifically tailored to a particular 3D representation (Chen et al., 2024b). To the best of our knowledge, DreamCatalyst is the first to achieve state-of-the-art results and provide extensive experiments on both NeRF and 3DGS editing. In summary, our key contributions are as follows:

- We suggest a general formulation for 3D editing by introducing a new interpretation of DDS as a reverse process of SDEdit.
- We present two conditions for designing objective function and a specialized formulation, which enable fast and high-quality 3D editing via controlling identity preservation and editability on both NeRF and 3DGS.
- We first introduce using FreeU for 3D editing to enhance editability to overcome the trade-offs inherent in reweighting terms of the formulation for editing objectives.

## 2 Preliminaries

### 2.1 Diffusion Models

A diffusion model (Song & Ermon, 2019; Ho et al., 2020) consists of a forward process that gradually perturbs a data point $x_0$ with a noise $\epsilon$ and a reverse process that progressively denoises the noisy data. The forward process is defined as $x_t = \sqrt{\bar{\alpha}_t}x_0 + \sqrt{1 - \bar{\alpha}_t}\epsilon, \epsilon \sim \mathcal{N}(0, \mathbf{I})$, where $\mathcal{N}(0, \mathbf{I})$ is a Gaussian distribution, $t \in [0, T]$ denotes the timestep, and $x_t$ represents perturbed $x_0$ at $t$. $\alpha_s$ and $\bar{\alpha}_t := \prod_{s=1}^{t} \alpha_s$ are predefined noise scheduling coefficients. In contrast, the reverse process utilizes the score function, which is predicted with a neural network, and a sampler. The score function is equivalent to the denoising network $\epsilon_{\theta}$, parameterized by $\theta$, of a diffusion model. It is trained via denoising score matching as $\min_{\theta} \mathcal{L}(x_0) = \mathbb{E}_{t,\epsilon}[\| \epsilon_{\theta}(x_t, t) - \epsilon \|_2^2]$.

**Denoising Diffusion Implicit Model (DDIM)**. In the context of DDIM (Song et al., 2020a), the reverse process can be expressed with Tweedie's formula (Efron, 2011) as

$$x_{t-1} = \sqrt{\bar{\alpha}_{t-1}}\hat{x}_{0|t} + \sqrt{1 - \bar{\alpha}_{t-1}}\tilde{\epsilon}, \tag{1}$$

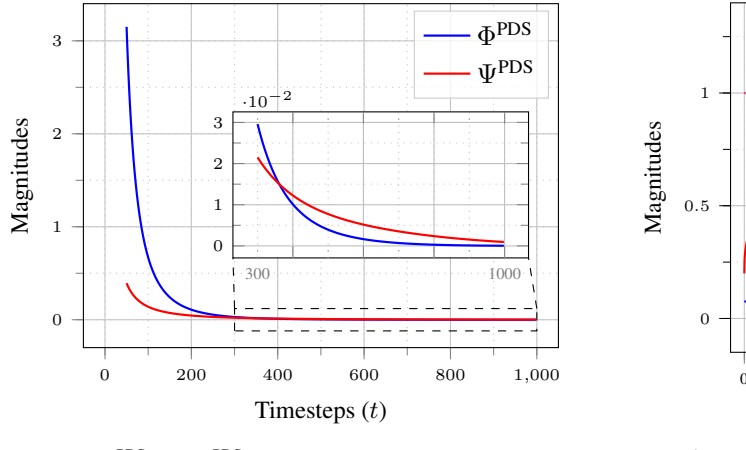 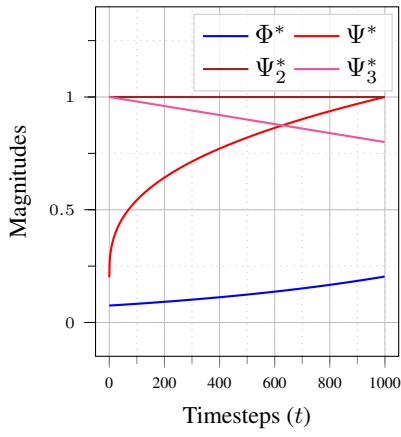

(a) $\Phi^{\text{PDS}}$ and $\Psi^{\text{PDS}}$ magnitudes over timesteps $t$      (b) $\Phi^*$ and $\Psi^*$ magnitudes over timesteps $t$

Figure 2: **Comparison of coefficients between PDS and our method across different timesteps.** We plot the weighting functions of PDS and DreamCatalyst in (a) and (b), respectively. $\Phi^{\text{PDS}}$ and $\Psi^{\text{PDS}}$ indicate the coefficient of identity preservation and editability of PDS. $\Phi^*$ and $\Psi^*$ are the coefficient of identity preservation and editability of ours, respectively. $\Psi_2^*$ and $\Psi_3^*$ are for extra special cases.

$$\text{where } \hat{\boldsymbol{x}}_{0|t} = (\boldsymbol{x}_t - \sqrt{1 - \bar{\alpha}_t}\boldsymbol{\epsilon}_\theta(\boldsymbol{x}_t, t))/\sqrt{\bar{\alpha}_t} \text{ and } \tilde{\boldsymbol{\epsilon}} = \frac{\sqrt{1 - \bar{\alpha}_{t-1} - \eta^2 \beta_t^2} \boldsymbol{\epsilon}_\theta(\boldsymbol{x}_t, t) + \eta\beta_t\boldsymbol{\epsilon}}{\sqrt{1 - \bar{\alpha}_{t-1}}}. \quad (2)$$

Here, $\hat{\boldsymbol{x}}_{0|t}$ is a predicted $\boldsymbol{x}_0$ with $\boldsymbol{x}_t$, and $\tilde{\boldsymbol{\epsilon}}$ is a noise term consisting of a deterministic term and a stochastic term $\boldsymbol{\epsilon} \sim \mathcal{N}(0, \mathbf{I})$. The deterministic sampling is achieved when $\eta\beta_t = 0$, as the stochasticity of noise term $\tilde{\boldsymbol{\epsilon}}$ can be manipulated with the stochastic property controlling a hyperparameter $\eta$ and a coefficient $\beta_t = \sqrt{(1 - \bar{\alpha}_{t-1})/(1 - \bar{\alpha}_t)}\sqrt{1 - \bar{\alpha}_t/\bar{\alpha}_{t-1}}$.

## 2.2 SCORE DISTILLATION SAMPLING

In this section, we review previous SDS methods that utilize pre-trained diffusion models as priors. Unlike the diffusion models that perform sampling in image space, DreamFusion (Poole et al., 2022) proposes the SDS framework that conducts sampling in the parameter space. SDS optimizes parameterized models, such as differentiable image generators like NeRF and 3DGS, using the diffusion training objective. In 3D generation, SDS perturbs rendered images $\boldsymbol{x} = g(\psi, c)$ with noise $\boldsymbol{\epsilon}$, where $g$ is a NeRF or 3DGS model, $\psi$ are parameters of the model $g$, and $c$ is a camera parameter. It is then distilled from a pre-trained diffusion model with rich 2D priors to train NeRF or 3DGS. The training objective is defined as follows:

$$\min_\psi \mathcal{L}_{\text{SDS}}(\boldsymbol{x}_0 = g(\psi, c)) = \mathbb{E}_{t,\boldsymbol{\epsilon}}[\|\boldsymbol{\epsilon}_\theta^\omega(\boldsymbol{x}_t, y, t) - \boldsymbol{\epsilon}\|_2^2], \quad (3)$$

where the predicted noise with classifier-free guidance (CFG) is $\boldsymbol{\epsilon}_\theta^\omega(\boldsymbol{x}_t, y, t) := \boldsymbol{\epsilon}_\theta(\boldsymbol{x}_t, y_\varnothing, t) + \omega_y(\boldsymbol{\epsilon}_\theta(\boldsymbol{x}_t, y, t) - \boldsymbol{\epsilon}_\theta(\boldsymbol{x}_t, y_\varnothing, t))$ (Ho & Salimans, 2022), $\omega_y$ indicates the scale of text-guidance, $y$ is a text prompt, and $y_\varnothing$ is a null-text. Particularly, SDS omits the U-Net Jacobian term for computation efficiency as $\nabla_\psi \mathcal{L}_{\text{SDS}}(\boldsymbol{x}_0 = g(\psi, c)) = \mathbb{E}_{t,\boldsymbol{\epsilon}}[(\boldsymbol{\epsilon}_\theta^\omega(\boldsymbol{x}_t, y, t) - \boldsymbol{\epsilon})\frac{\partial \boldsymbol{x}_0}{\partial \psi}]$.

Even though SDS is an effective framework for 3D generation, SDS suffers from editing tasks since SDS does not reflect identity preservation.

**Delta Denoising Score (DDS).** The editing task consists of two key aspects: (1) preserving the source content's identity and (2) aligning with the target text prompt. Since the objective of SDS is designed for the generation task, it struggles to preserve the source identity. To address this, DDS is proposed to preserve the source identity by modifying the SDS loss function, as defined below:

$$\mathcal{L}_{\text{DDS}}(\boldsymbol{x}_0^{\text{tgt}} = g(\psi, c)) = \mathbb{E}_{t,\boldsymbol{\epsilon}}[\|\boldsymbol{\epsilon}_\theta^\omega(\boldsymbol{x}_t^{\text{tgt}}, y^{\text{tgt}}, t) - \boldsymbol{\epsilon}_\theta^\omega(\boldsymbol{x}_t^{\text{src}}, y^{\text{src}}, t)\|_2^2], \quad (4)$$

where $\boldsymbol{x}_t^{\text{tgt}}$ and $\boldsymbol{x}_t^{\text{src}}$ are the perturbed target data $\boldsymbol{x}_0^{\text{tgt}}$ and source data $\boldsymbol{x}_0^{\text{src}}$ at a timestep $t$ with a random noise $\boldsymbol{\epsilon}$, and $y^{\text{tgt}}$ and $y^{\text{src}}$ are the target prompt and source prompt, respectively. The estimated noise of $\boldsymbol{x}_t^{\text{src}}$ becomes a pivot for maintaining the source identity.

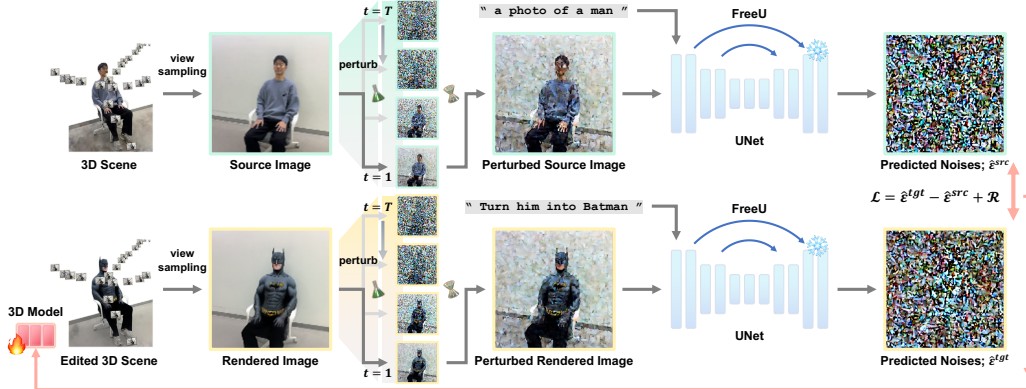

Figure 3: **Overall architecture.** DreamCatalyst approximates inversion-based SDEdit with DDS loss and identity regularizer $\mathcal{R}_{\text{iden}}$. Furthermore, DreamCatalyst utilizes FreeU to enhance 3D editing quality without additional computational cost and memory usage.

**Posterior Distillation Sampling.** We focus on SDS-based editing, a direct 3D editing method that achieves 3D-consistent results. It is contrast to Instruct-Nerf2NeRF (IN2N) (Haque et al., 2023), which edits source scenes in 2D space. While DDS shows remarkable editing in 2D images, it suffers from quality degradation in 3D editing. This degradation occurs because 3D editing requires stronger identity preservation than 2D editing. However, DDS optimizes NeRF by minimizing the residual between the SDS loss of the source and target without any additional regularization for identity preservation. The absence of regularization leads to deviation from the source. To address this, PDS introduces a stochastic latent matching loss to add an explicit identity preservation term in DDS loss. The stochastic latent $z_t$, which contains the structural details of $x_0$, is calculated as $z_t(x_0, y) = (x_{t-1} - \mu_\theta(x_t, y))/\sigma_t$, where $\sigma_t := \frac{1-\bar{\alpha}_{t-1}}{1-\bar{\alpha}_t}(1 - \alpha_t)$ and $\mu_\theta(x_t, y) = (\sqrt{\bar{\alpha}_{t-1}}(1 - \alpha_t)\hat{x}_{0|t} + \sqrt{\alpha_t}(1 - \bar{\alpha}_{t-1})x_t)/(1 - \bar{\alpha}_t)$. Therefore, the stochastic latent matching loss is as follows:

$$\mathcal{L}_{\text{PDS}}(x_0^{\text{tgt}} = g(\psi, c)) = \mathbb{E}_{t,\epsilon}[\| z_t(x_t^{\text{tgt}}, y^{\text{tgt}}) - z_t(x_t^{\text{src}}, y^{\text{src}}) \|_2^2] \qquad (5)$$

By ignoring the U-Net Jacobian term as done in SDS, the gradient of stochastic latent matching loss $\mathcal{L}_{\text{PDS}}$ is written as

$$\nabla_\psi \mathcal{L}_{\text{PDS}} = \mathbb{E}_{t,\epsilon}[(z_t(x_t^{\text{tgt}}, y^{\text{tgt}}) - z_t(x_t^{\text{src}}, y^{\text{src}}))\frac{\partial x_0^{\text{tgt}}}{\partial \psi}] \qquad (6)$$

$$= \mathbb{E}_{t,\epsilon}[(\Phi^{\text{PDS}}(t)\underbrace{(x_0^{\text{tgt}} - x_0^{\text{src}})}_{\text{identity preservation}} + \Psi^{\text{PDS}}(t)\underbrace{(\epsilon_\theta^\omega(x_t^{\text{tgt}}, y^{\text{tgt}}, t) - \epsilon_\theta^\omega(x_t^{\text{src}}, y^{\text{src}}, t))}_{\text{gradient of } \mathcal{L}_{\text{DDS}}})\frac{\partial x_0^{\text{tgt}}}{\partial \psi}]. \quad (7)$$

$\Phi^{\text{PDS}}(t)$ and $\Psi^{\text{PDS}}(t)$ are the defined coefficients as a function of the timestep $t$, each representing the coefficient of identity preservation and that of editability, respectively. The gradient of stochastic latent matching is equivalent to Equation 7, which means the PDS loss implicitly involves the explicit identity preservation term and DDS gradient term for editing.

## 3 DREAMCATALYST

### 3.1 MOTIVATION

We aim to design an objective function, like PDS, possessing two important properties. (1) It should include an explicit term for strong identity preservation. (2) It should align with the roles of diffusion timesteps and reduce the conflict with the approximated diffusion reverse sampling. To achieve this goal, identity preservation has to be stressed in large noise perturbation and does not diverge in small levels of perturbation by reweighting each term of Equation 7. However, the nature of the formulation of stochastic latent matching implicitly includes an identity preservation term and the gradient of the DDS loss, making it incapable of directly adjusting the coefficients. Therefore, we provide a new interpretation of DDS and introduce a general formulation of PDS through this perspective to

reweigh the terms. Furthermore, we propose a specialized formulation that aligns with the diffusion timestep roles and supports the diffusion reverse process. The specialized formulation has mainly two advantages: (1) our formulation leads to fine-detailed 3D edited results by considering diffusion timestep roles and (2) it immensely reduces training time with the diffusion reverse process.

## 3.2 GENERAL FORMULATION OF PDS

In this section, we unveil the relationship between the reverse SDEdit process and DDS (see Appendix A for preliminaries on SDEdit). The key insight of DreamCatalyst is that *the objective of DDS is equivalent to the single-step DDIM-based SDEdit sampling.* The reverse SDEdit process enables stochastic editing by solving the stochastic differential equations (SDEs) (Song et al., 2020b) with random sampled noise as $\boldsymbol{x}_{t-1} = \sqrt{\bar{\alpha}_{t-1}}\hat{\boldsymbol{x}}_{0|t} + \sqrt{1-\bar{\alpha}_{t-1}}\boldsymbol{\epsilon}, \boldsymbol{\epsilon} \sim \mathcal{N}(0, \mathbf{I})$. However, recent editing studies (Tumanyan et al., 2023; Cao et al., 2023) utilize the DDIM inversion to preserve the source identity. By combining the SDEdit and DDIM inversion for identity preservation, the DDIM-based SDEdit sampling is defined as

$$\boldsymbol{x}_{t-1}^{\text{tgt}} = \sqrt{\bar{\alpha}_{t-1}}\hat{\boldsymbol{x}}_{0|t}^{\text{tgt}} + \sqrt{1-\bar{\alpha}_{t-1}}\tilde{\boldsymbol{\epsilon}}, \tag{8}$$

where $\boldsymbol{x}_0^{\text{tgt}}$ indicates the rendered image to edit. Especially when we define $\eta\beta_t = 0$ for deterministic sampling, the noise is as $\tilde{\boldsymbol{\epsilon}} = \boldsymbol{\epsilon}_\theta(x_{t-1}^{\text{src}}, y^{\text{src}}, t)$. In this case, DDIM inversion-based perturbed image is defined as $\boldsymbol{x}_t^{\text{tgt}} = \sqrt{\bar{\alpha}_t}\boldsymbol{x}_0^{\text{tgt}} + \sqrt{1-\bar{\alpha}_t}\boldsymbol{\epsilon}_\theta(\boldsymbol{x}_t^{\text{src}}, y^{\text{src}}, t)$, which is the result of applying a single forward step. We can rewrite the Equation 8 with Equation 2 and the forward step as follows:

$$\boldsymbol{x}_{t-1}^{\text{tgt}} = \sqrt{\bar{\alpha}_{t-1}}\left(\frac{\boldsymbol{x}_t^{\text{tgt}}}{\sqrt{\bar{\alpha}_t}} - \frac{\sqrt{1-\bar{\alpha}_t}}{\sqrt{\bar{\alpha}_t}}\boldsymbol{\epsilon}_\theta^\omega(\boldsymbol{x}_t^{\text{tgt}}, y^{\text{tgt}}, t)\right) + \sqrt{1-\bar{\alpha}_{t-1}}\tilde{\boldsymbol{\epsilon}} \tag{9}$$

$$= \sqrt{\bar{\alpha}_{t-1}}\left(\boldsymbol{x}_0^{\text{tgt}} + \frac{\sqrt{1-\bar{\alpha}_t}}{\sqrt{\bar{\alpha}_t}}\boldsymbol{\epsilon}_\theta^\omega(\boldsymbol{x}_t^{\text{src}}, y^{\text{src}}, t) - \frac{\sqrt{1-\bar{\alpha}_t}}{\sqrt{\bar{\alpha}_t}}\boldsymbol{\epsilon}_\theta^\omega(\boldsymbol{x}_t^{\text{tgt}}, y^{\text{tgt}}, t)\right) + \sqrt{1-\bar{\alpha}_{t-1}}\tilde{\boldsymbol{\epsilon}}. \tag{10}$$

Although the single-step denoising process of SDEdit is clear in the diffusion process with Equation 8, inspired by DreamSampler (Kim et al., 2024), we can interpret the process as an optimization problem as follows:

$$\boldsymbol{x}_{t-1}^{\text{tgt}} = \sqrt{\bar{\alpha}_{t-1}}\bar{\boldsymbol{x}} + \sqrt{1-\bar{\alpha}_{t-1}}\tilde{\boldsymbol{\epsilon}}, \tag{11}$$

$$\bar{\boldsymbol{x}} = \arg\min_{\boldsymbol{x}_0^{\text{tgt}}} \| \hat{\boldsymbol{x}}_{0|t}^{\text{tgt}} - \boldsymbol{x}_0^{\text{tgt}} \|^2 = \arg\min_{\boldsymbol{x}_0^{\text{tgt}}} \frac{\sqrt{1-\bar{\alpha}_t}}{\sqrt{\bar{\alpha}_t}} \| \boldsymbol{\epsilon}_\theta^\omega(\boldsymbol{x}_t^{\text{tgt}}, y^{\text{tgt}}, t) - \boldsymbol{\epsilon}_\theta^\omega(\boldsymbol{x}_t^{\text{src}}, y^{\text{src}}, t) \|^2. \tag{12}$$

Equation 12 indicates that the DDS objective is equivalent to the objective of the optimization problem, when $\boldsymbol{x}_0^{\text{tgt}} = g(\psi, c)$. Thus, solving the DDS objective ensures equivalence to the single-step process of SDEdit. By extension, *optimizing the DDS objective with decreasing timestep sampling corresponds to the reverse SDEdit process.* We notice that the proposed inversion is a proximal inversion. Conventional DDIM-inversion calculates $\tilde{\boldsymbol{\epsilon}}$ with a multi-step inversion for pivoting. However, this multi-step inversion method requires extensive calculations for each multi-view image in 3D editing. To mitigate the computational burden, our method samples a single-step $\tilde{\boldsymbol{\epsilon}}$ for each level of noise perturbation, which enables the proximal inversion due to different $\tilde{\boldsymbol{\epsilon}}$ with respect to $t$. Moreover, we stress that DreamSampler assumes $\boldsymbol{x}_0^{\text{tgt}} = \boldsymbol{x}_0^{\text{src}}$, despite introducing a similar observation. This assumption differs from the formulation presented in DDS (Hertz et al., 2023). Therefore, under this assumption, the general DDS objective presented in Equation 4 cannot be interpreted as an optimization problem. In contrast, we provide a more general interpretation of DDS that is not limited by this assumption, offering a broader perspective compared to DreamSampler.

Since we express the DDS objective as the optimization problem as in Equation 12, we can apply additional regularization terms for enforcing identity preservation. These regularization terms allow supplementary guidance during the reverse SDEdit process, addressing the lack of identity preservation in DDS. Therefore, we propose the general formulation of PDS by adding the identity preservation regularizer $\mathcal{R}_{\text{iden}}$ to the DDS objective. That is,

$$\mathcal{L}_{\text{iden}}(\boldsymbol{x}_0^{\text{tgt}} = g(\psi, c)) = \mathbb{E}_{t, \boldsymbol{\epsilon}}[\Phi(t)\mathcal{R}_{\text{Iden}} + \Psi(t)\mathcal{L}_{\text{DDS}}], \quad \text{where} \quad \mathcal{R}_{\text{Iden}} = \| \boldsymbol{x}_0^{\text{tgt}} - \boldsymbol{x}_0^{\text{src}} \|_2^2, \tag{13}$$

and $\Phi(t)$ and $\Psi(t)$ are weighting functions of the general formulation. Herein, the regularizer $\mathcal{R}_{\text{Iden}}$ preserves the identity and DDS loss $\mathcal{L}_{\text{DDS}}$ takes charge of editing. By understanding the explicit role of each term in our loss function, we can easily control the formulation of PDS with Equation 13. With this controllable, generalized formulation of PDS, we then develop a specialized formulation that accounts for the sampling dynamics of diffusion models in the following section.

### 3.3 Diffusion-friendly SDS-based Editing

Since our generalized PDS formula allows for explicit control of each term at every timestep, we can fully leverage the controllability to align the 3D editing process with the sampling dynamics of diffusion models. Specifically, we propose a specialized formulation of Equation 13, which considers the roles of diffusion timestep and their alignment with the reverse SDEdit process. The design choice of formulation in DreamCatalyst aims to satisfy two conditions: (1) strong identity preservation in large timesteps and (2) reducing identity preservation in small timesteps. The first condition, strong identity preservation in large timesteps, reduces the information loss of source features in high levels of noise perturbation. This condition enables the preservation of the source features in the early diffusion reverse process. The second condition, weak identity preservation in small timesteps, leads to synthesizing fine details for 3D editing as the role of diffusion. The proposed specialized formulation of DreamCatalyst, which satisfies two conditions, is as follows:

$$\mathcal{L}_{\text{DreamCatalyst}}(\boldsymbol{x}_0^{\text{tgt}} = g(\psi, c)) = \mathbb{E}_{t,\boldsymbol{\epsilon}}[\Phi^*(t)\mathcal{R}_{\text{Iden}} + \Psi^*(t)\mathcal{L}_{\text{DDS}}], \tag{14}$$

$$\Phi^*(t) = \chi e^{t/T}, \Psi^*(t) = \delta + \gamma\sqrt[e]{t/T}, \tag{15}$$

and $\chi$, $\delta$, $\gamma$ are hyperparameters, respectively. We set $\chi = 0.075$, $\delta = 0.2$, and $\gamma = 0.8$ for all experiments. As shown in Fig. 2b, the formulation of DreamCatalyst fulfills the two conditions, thereby our objective function is suitable for editing tasks and diffusion reverse process. The primary difference between DreamCatalyst and PDS lies in the formulation of $\Phi(t)$ and $\Psi(t)$. This modification reduces editing time in two key ways: (1) the reweighting scheme enables the use of the approximated diffusion reverse process, familiar to the diffusion sampling procedure; (2) DreamCatalyst avoids inefficient distillation. In PDS, distillation at small timesteps leads to excessive identity preservation, which disrupts editing. In contrast, our weighting facilitates efficient distillation without interrupting the editing process. These factors significantly reduce the number of optimization steps, thereby decreasing the overall editing time.

The SDEdit process with minimizing $\mathcal{L}_{\text{DreamCatalyst}}$ requires diffusion reverse process-likely timestep sampling. To achieve this, we adopt decreasing timestep sampling, which uniformly samples timestep $t = T \rightarrow 1$. While the non-increasing timestep sampling (Huang et al., 2023) is also a good option, we employ decreasing timestep sampling for fulfilling Equation 12 in all timesteps. Our specialized objective function with decreasing timestep sampling enables the SDEdit process with a parameterized model, especially NeRF and 3DGS in this paper. The overall framework of DreamCatalyst is illustrated in Fig. 3. Initially, $t$ is sampled with decreasing timestep sampling. Subsequently, both the rendered and source images are perturbed according to $t$. Equation 14 is then computed with each perturbed image. Finally, the 3D model is optimized by the loss function. We omit the U-Net Jacobian term as previous works to calculate the gradient of $\mathcal{L}_{\text{DreamCatalyst}}$.

We demonstrate that fulfilling the two conditions enables effective 3D editing as Fig. 1. Dream-Catalyst not only achieves high-quality 3D scene edits across various scenarios but also performs editing faster than existing methods. Besides, we present two more special cases and show that the coefficients of specialized formulation remain robust as long as the two conditions are satisfied in Fig. 2b. For an intuitive comparison, we fix $\Phi^*$ and vary only $\Psi^*$, setting the two additional cases as $\Psi_2^*(t) = 1$ and $\Psi_3^*(t) = 1 - 0.2t/T$. In section 4, we demonstrate that these special cases, which satisfy the two conditions, also surpass the existing state-of-the-art baselines. Thus, fulfilling two conditions enhances editing speed and quality.

### 3.4 Enhancing editability with FreeU

For an improved architecture, we introduce utilizing FreeU in 3D editing to enhance editability without additional memory usage and computational costs. FreeU suppresses high-frequency features by scaling up the backbone features, which contain a large amount of low-frequency information (Si et al., 2023). The amplifying backbone features stresses low-frequency features, thereby relatively reducing the impact of high-frequency features. Consequently, suppressing high-frequency features increases editability as sharp characteristics of high-frequency features are smoothed as the edge features are weakened. Moreover, identity preservation, corresponding to the low-frequency domain, is maintained by amplifying backbone features. In conclusion, FreeU enhances editability without compromising identity preservation. In addition, adopting FreeU instead of Dreambooth and LoRA removes additional module computations. Consequently, integrating FreeU reduces editing time without incurring extra computational overhead.

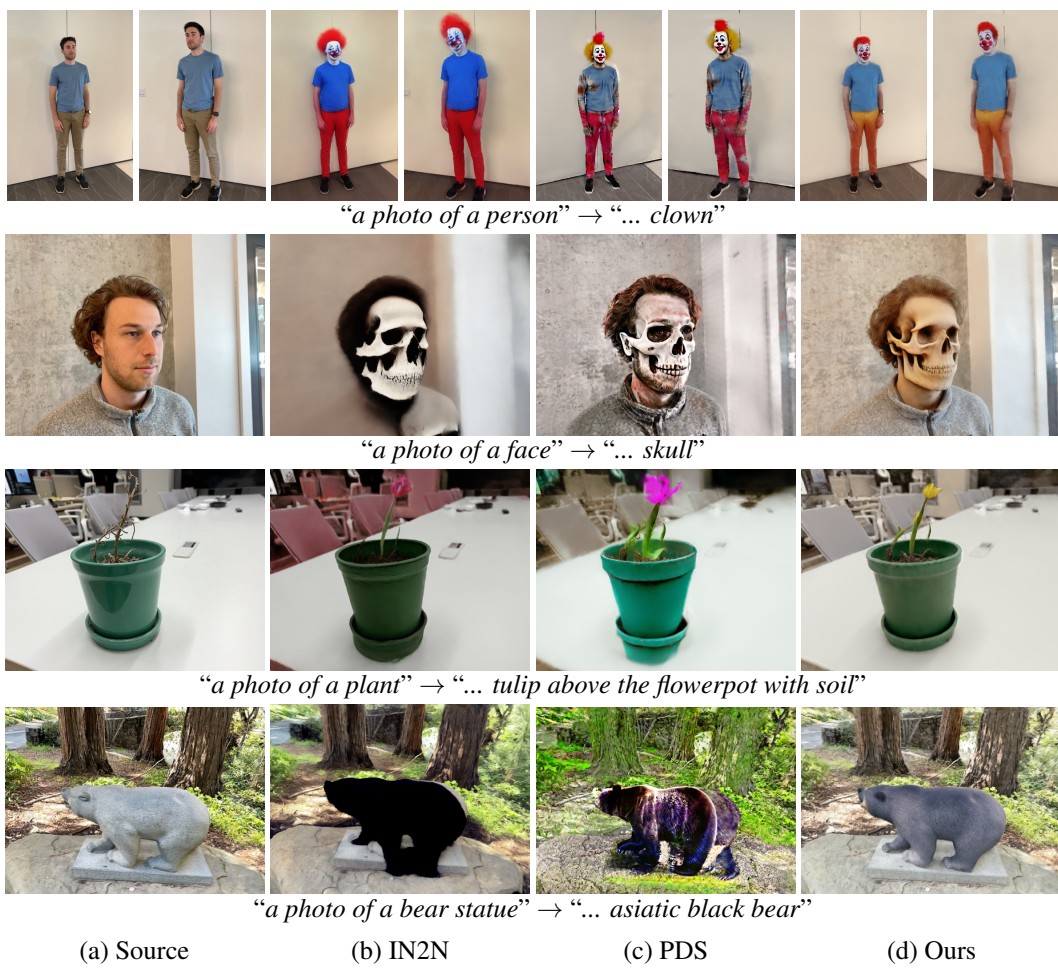

"*a photo of a person*" → "*... clown*"

"*a photo of a face*" → "*... skull*"

"*a photo of a plant*" → "*... tulip above the flowerpot with soil*"

"*a photo of a bear statue*" → "*... asiatic black bear*"

(a) Source        (b) IN2N        (c) PDS        (d) Ours

Figure 4: **Qualitative comparison with baseline methods on NeRF scenes.** We provide visual editing results in NeRF scenes for each baseline method and DreamCatalyst. DreamCatalyst demonstrates more photorealistic editability while preserving the identity of source scenes such as structures and backgrounds.

## 4 EXPERIMENTS

We conduct experiments on real scenes using datasets from IN2N and PDS. The types of scenes include a *sitting person, a full-body person, a face, objects, and outdoor* scenes. We evaluate our method and baselines in eight scenes with 40 pairs of source and target text prompts. For comparisons, we evaluate our method against state-of-the-art baselines on NeRF scenes: IN2N and PDS, as well as on 3DGS scenes: PDS, GaussianEditor (Chen et al., 2024b) and DGE (Chen et al., 2024a). Furthermore, we compare two modes of DreamCatalyst: (1) a high-quality mode, and (2) a fast mode, which requires fewer training iterations than the high-quality mode. Additionally, we present an ablation study addressing how FreeU affects DreamCatalyst. Unless explicitly stated as using $\Psi_2^*(t)$ or $\Psi_3^*(t)$, all results presented from our method (in both NeRF and 3DGS scenes) are based on the use of the default $\Psi^*(t)$ coefficient.

### 4.1 QUALITATIVE EVALUATION

In Fig. 4, we present a qualitative comparison with the baseline methods on NeRF scenes, focusing on both identity preservation and editing effectiveness. DreamCatalyst consistently generates more detailed and photorealistic results compared to the baseline methods (e.g., tulips generated by other methods appear blurred and lack fine details, compromising the edit). Furthermore, while other methods result in blurry and over-saturated backgrounds, DreamCatalyst preserves the identity of the source scenes by maintaining background details and overall structure. Although PDS aligns well with the target text prompt in terms of subject editing, it struggles with identity preservation.

Table 1: **Quantitative comparison on NeRF scenes.** Ours outperforms the baseline methods on NeRF editing. **Bold** represents the best result, and underline indicates the second-best result.

| Method | CLIP-Direc (↑) | CLIP-Img (↑) | Aesthetic (↑) | Total time (min, ↓) |
|---|---|---|---|---|
| IN2N | 0.157 | 0.722 | 5.399 | ∼ 130 |
| PDS | 0.161 | 0.687 | 5.437 | ∼ 580 |
| Ours (fast) | 0.165 | 0.746 | 5.557 | ∼ **25** |
| Ours | **0.180** | 0.746 | **5.688** | ∼ 70 |
| Ours (w/ $\Psi_2^*$) | 0.178 | 0.745 | 5.659 | ∼ 70 |
| Ours (w/ $\Psi_3^*$) | 0.178 | **0.749** | 5.659 | ∼ 70 |

Table 2: **Quantitative comparison on 3DGS scenes. Bold** represents the best result in each category, and underline indicates the second-best result among the model-specific methods.

| Category | Method | CLIP-Direc (↑) | CLIP-Img (↑) | Aesthetic (↑) |
|---|---|---|---|---|
| Model-Agnostic | PDS | 0.108 | 0.627 | 4.977 |
| | Ours | **0.171** | **0.724** | **5.336** |
| Model-Specific | GE (DDS) | 0.150 | 0.593 | 4.860 |
| | GE (IN2N) | 0.138 | 0.765 | 5.508 |
| | DGE | **0.154** | 0.758 | 5.517 |
| | GE (Ours) | 0.152 | 0.773 | **5.588** |
| | GE (Ours+fast) | 0.143 | **0.777** | 5.541 |

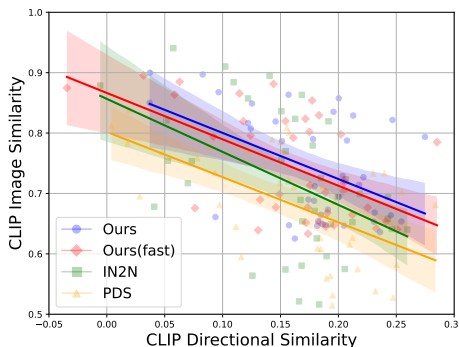

Figure 5: **Scatter plot comparing our method against baseline methods on NeRF scenes.** The plot shows NeRF editing performance on CLIP directional similarity and CLIP image similarity for baseline methods: IN2N, PDS, and our method, including the fast training mode (fast). Trend lines are fitted using linear regression. Shaded areas around the trend lines indicate the 95% confidence intervals.

Specifically, the backgrounds in PDS results are particularly prone to over-saturation or unrealistic color shifts. This highlights a key limitation of PDS, as it tends to prioritize the edit at the expense of maintaining the original scene's identity. In contrast, DreamCatalyst achieves a better balance editability and identity preservation, resulting in superior overall performance. Qualitative comparisons on 3DGS scenes are provided in Appendix C.3.

## 4.2 QUANTITATIVE EVALUATION

We evaluate DreamCatalyst and baseline methods using three key metrics: CLIP directional similarity (Patashnik et al., 2021), CLIP image similarity, and aesthetic score (Schuhmann, 2022). CLIP directional similarity measures image-text alignment, CLIP image similarity assesses identity preservation, and the aesthetic score reflects the visual quality of the edits. As shown in Tab. 1 and Fig. 5, DreamCatalyst achieves the highest scores across all metrics on NeRF editing. This is particularly notable, as other baseline methods, such as PDS, tend to excel in one area—such as high CLIP directional similarity or aesthetic score—but struggle in others, particularly in maintaining identity preservation. DreamCatalyst, however, strikes a balance across all three metrics, generating edits that are both photorealistic and faithful to the source scene.

Additionally, we measure the editing time for each method. For a fair comparison, all methods are evaluated at the same resolution. DreamCatalyst with a fast mode is approximately 23 times faster than PDS, and the high-quality mode is about eight times faster than PDS. Despite IN2N performing edits in 2D space, which requires less time than direct 3D editing methods, DreamCatalyst is still $1.85\times$ faster than IN2N, even in the high-quality mode. Moreover, as shown in Tab. 1, the results from our method using $\Psi_2^*$ and $\Psi_3^*$ consistently outperform the baselines across all metrics.

In the 3DGS setting, we first compare DreamCatalyst with baseline model-agnostic editing method, which are distinct from 3DGS-specific approaches. Further details of model-agnostic and model-specific methods are in Appendix B.2. As demonstrated in Tab. 2, DreamCatalyst achieves the highest scores across most metrics on 3DGS editing. This indicates that our approach is an effective model-agnostic editing method applicable to both NeRF and 3DGS scenes. Additionally, integrating DreamCatalyst with GaussianEditor (GE) yields state-of-the-art results, demonstrating that DreamCatalyst enhances the performance of 3DGS-specific methods. Furthermore, the fast mode of GE with DreamCatalyst not only outperforms the baselines but also achieves a training speedup of $1.25\times$ compared to the vanilla GE. We emphasize that as better 3DGS baseline architectures using score distillation emerge, our approach can be applied to further improve their performance.

Table 3: **User studies.** We conduct user studies to measure human preference across three criteria. Our method is more preferred than other baselines. **Bold** indicates the best result.

| Method | Prompt Alignment ($\uparrow$) | Overall Quality ($\uparrow$) | Identity Preservation ($\uparrow$) |
|---|---|---|---|
| IN2N | 19.13% | 20.08% | 20.05% |
| PDS | 22.58% | 19.21% | 20.21% |
| Ours | **58.29%** | **60.71%** | **59.74%** |

Table 4: **Ablation on FreeU.** Quantitative ablation study on the effects of FreeU. Using FreeU with a setting of $b = 1.1$ achieves a balanced trade-off between editability and identity preservation in the generated results. **Bold** indicates the best result.

| $b$ | CLIP-Direc ($\uparrow$) | CLIP-Img ($\uparrow$) | Aesthetic ($\uparrow$) |
|---|---|---|---|
| 1.0 (w/o FreeU) | 0.171 | 0.744 | 5.564 |
| 1.1 (ours) | 0.180 | **0.746** | **5.688** |
| 1.3 | **0.183** | 0.710 | 5.624 |

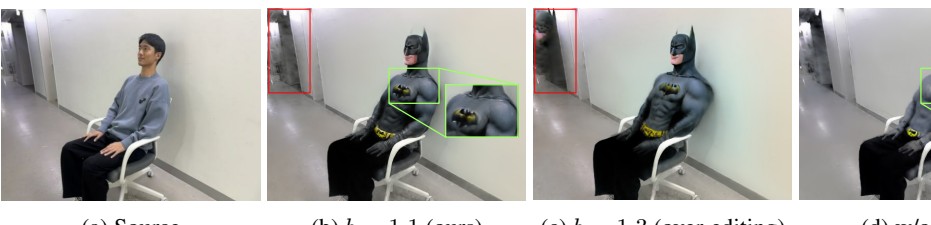

(a) Source      (b) $b = 1.1$ (ours)      (c) $b = 1.3$ (over-editing)      (d) w/o FreeU

Figure 6: **Qualitative ablation on FreeU.** The first row shows the ablation of FreeU with the text prompt "Turn him into Batman." The use of FreeU enhances editability, as demonstrated by the Batman logo (green bounding box) and instances of over-editing (red bounding box).

### 4.3 USER STUDY

We conduct user studies, as in Tab. 3, because the metrics assessing 2D images are insufficient for evaluating 3D scenes. Participants were asked to select the best video from the baselines and DreamCatalyst based on 15 text prompts, evaluated across three criteria: (1) prompt alignment, (2) overall quality, and (3) identity preservation. To gather human preference data, we utilized Amazon Mechanical Turk to survey 100 participants. As a result, DreamCatalyst is preferred over the baselines by a large margin across all criteria, receiving nearly three times as many selections compared to other models. Further details about the user studies are provided in Appendix E.3.

### 4.4 ABLATIONS

We demonstrate the effectiveness of FreeU in our method with qualitative and quantitative comparisons. FreeU modifies the scale of upsampling features in U-Net decoder using a parameter $b$. In this framework, increasing the value of $b$ leads to the suppression of high-frequency components in an image. We hypothesize that this characteristic of FreeU facilitates easier editing. However, if $b$ is set too high, the editing process becomes excessively easy. This hypothesis is supported by the results shown in Fig. 6 (b) and (c), where the use of FreeU with $b = 1.3$ results in excessive editing. This over-editing is not confined to the primary subject but extends to the background as well. While increasing $b$ in FreeU can enhance the editing process, excessive suppression of high-frequency components can lead to overly smooth results and unintended editing artifacts. As shown in Tab. 4, despite CLIP directional similarity increases as $b$ increases, the values of the other metrics are decreased in $b = 1.3$. Thus, FreeU with a setting of $b = 1.1$ achieves balanced results.

### 5 CONCLUSION

We propose a general formulation for 3D editing by unveiling the relationship between the reverse SDEdit process and DDS. Based on this formulation, we introduce DreamCatalyst, which considers the dynamics of a diffusion process to edit 3D scenes with an SDS-based approach as a reverse SDEdit process. Moreover, we suggest using FreeU in Score Distillation to overcome the trade-offs between editability and identity preservation inherent in the formulation. Consequently, DreamCatalyst achieves fast and high-quality 3D editing on both NeRF and 3DGS scenes. Through comparative analysis and user studies, we demonstrate that DreamCatalyst surpasses state-of-the-art methods in both performance and editing speed.

## Acknowledgements

This research was supported by the Basic Science Research Program through the National Research Foundation of Korea (NRF) funded by the MSIP (NRF-2022R1A2C3011154, RS-2023-00219019) and the Ministry of Education (No. RS-2024-00348396), Institute of Information & communications Technology Planning & Evaluation (IITP) grant funded by the Korea government (MSIT) (RS-2019-II190075, Artificial Intelligence Graduate School Program(KAIST), No. 2021-0-02068, Artificial Intelligence Innovation Hub, and No. RS-2024-00457882, National AI Research Lab Project), and KEIT grant funded by the Korean government (MOTIE) (No. 2022-0-00680, No. 2022-0-01045).

## Ethics Statement

There have been several advancements in text-driven 3D scene editing, yet few methods (Koo et al., 2023) focus on altering the formulation of score distillation sampling to preserve the identity of the original scene. Our method enhances control over the degree of identity preservation, allowing for more careful and targeted edits that prevent excessive deviation from the source scene. This controllability is particularly useful in applications where the integrity of the original 3D scene must be maintained.

However, the ability to precisely manipulate 3D content brings with it significant ethical responsibilities. We emphasize the need for robust ethical guidelines and regulations to prevent the misuse of this technology that can violate human rights. The potential for misuse underscores the importance of ongoing discussions about the ethical frameworks.

## Reproducibility Statement

Please refer to our official repository: https://github.com/kaist-cvml/DreamCatalyst. This repository provides the source code, setup, and datasets used in the experiments.

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

## A    PRELIMINARIES

**SDEdit.** SDEdit (Meng et al., 2021) is proposed to edit the images with the trained score network (i.e., a denoising network $\epsilon_\theta$ in diffusion models) by solving the stochastic differential equations (SDEs) (Song et al., 2020b). SDEdit moves the data point sequentially by leveraging the estimated score function $\nabla \log p(\boldsymbol{x}_t) \approx \lambda \epsilon_\theta(\boldsymbol{x}_t, t)$ with the reverse Variance Preserving (VP) SDE:

$$\boldsymbol{x}_{t-1} = \sqrt{\bar{\alpha}_{t-1}}\hat{\boldsymbol{x}}_{0|t} + \sqrt{1 - \bar{\alpha}_{t-1}}\boldsymbol{\epsilon}, \boldsymbol{\epsilon} \sim \mathcal{N}(0, \mathbf{I}), \tag{16}$$

where $\lambda$ is a scaling factor. Iteratively shifting the data point with the reverse VP-SDE synthesizes the edited image.

## B    IMPLEMENTATION DETAILS

### B.1    NERF EDITING

We utilize Nerfstudio (Tancik et al., 2023) for experiments. We train the `nerfacto` model from Nerfstudio for initialization with source scenes. For a fair comparison, we set different training steps for each method: 3,000 iterations for our method, 1,000 iterations for our method with the fast mode, 15,000 iterations for IN2N, and 30,000 iterations for PDS, following the original settings of their respective baselines. Based on previous researches (Poole et al., 2022; Katzir et al., 2023) indicating that higher classifier guidance may lead to over-saturated and poor edited results, we select a weight of 7.5 for classifier-free guidance in our method. During editing, we train the model using the Adam optimizer and an exponential decay learning rate scheduler. Specifically, since our method requires fewer iterations than PDS, we set smaller warm-up steps of learning rate schedulers: 100 for proposal networks and fields, and 300 for camera optimizers. We use a DDIM scheduler with 500 inference steps.

We employ InstructPix2Pix (IP2P) (Brooks et al., 2023) as a pretrained diffusion model, which takes the source image as input to incorporate a prior of the source. For this reason, our method does not require additional training or fine-tuning, in contrast to PDS, which necessitates fine-tuning diffusion models with DreamBooth (Ruiz et al., 2023) in advance. Additionally, we do not apply the refinement stage on PDS for an impartial comparison. All experiments are conducted on a single NVIDIA A6000 GPU.

### B.2    3DGS EDITING

For a fair comparison in 3DGS editing, we classify the baseline methods into two categories: model-agnostic and model-specific approaches. Model-specific methods, such as Gaussian Editor (GE) (Chen et al., 2024b) and DGE (Chen et al., 2024a), are designed for 3DGS scene editing and are not directly applicable to NeRF editing. Specifically, GE introduces semantic tracing to enable continuous tracking of Gaussian semantic labels, dynamically constraining the editing regions. Additionally, it implements hierarchical Gaussian splatting, which categorizes Gaussians into different generations to mitigate the effects of stochastic generative guidance. DGE performs 3DGS editing by applying key-view edits in 2D space and propagating the edited features across other views using epipolar constraints. This method involves iterative dataset updates, where the 3D scene is edited based on the 2D-edited frames. However, the techniques proposed in DGE are designed for explicit 3D models and cannot be applied to NeRF scenes. In contrast, model-agnostic approaches like PDS and DreamCatalyst can be applied across different 3D editing frameworks including NeRF and 3DGS. These methods modify score distillation sampling formulations or diffusion architectures, making them independent of the specific 3D model used.

We utilize two frameworks for a fair comparison in 3DGS editing within each category: Nerfstudio for model-agnostic methods and Threestudio (Guo et al., 2023) for model-specific methods. This selection is based on the fact that PDS is based on Nerfstudio, while GE and DGE employ Threestudio. For model-agnostic methods, we train the `splatfacto` model from Nerfstudio for initialization with source scenes, following the setup in the PDS paper. We set 3,000 training steps for our method and 30,000 steps for PDS, adhering to the PDS baseline settings. For model-specific methods, we choose 1,500 steps for our method with GE in high-quality mode and 1,200 steps in fast mode. Additionally, we set 500 steps and 1,500 steps for DGE and GE respectively, following

the original settings of their baselines. As detailed in Appendix B.1 for NeRF Editing, we use the same small warm-up steps for the learning rate schedulers for Gaussian parameters in Nerfstudio, including centers, spherical harmonics, opacity, scaling, and rotation. These parameters, along with camera parameters, are trained for a maximum of 3,000 steps. In the comparison of model-specific methods using Threestudio, we apply the same learning rate scheduler settings for Gaussian parameters. Additionally, to prevent over-densification in GE, we limit densification phases to 1,200 iterations in the high-quality mode of our method, ensuring that excessive cloning or splitting does not occur at the end of the editing process. All other parameters follow the original GE settings, and the experiments are conducted on a single NVIDIA A6000 GPU.

### B.3 TEXT-GUIDANCE IN DREAMCATALYST

In DreamCatalyst, we employed InstructPix2Pix (IP2P), which is prevalently used in NeRF and 3D Gaussian Splatting editing (Kim et al., 2023; Palandra et al., 2024), for instructive editing. The guidance of IP2P is composed of image and text conditioning. DreamCatalyst sets $\omega_y = 0$ for $\epsilon_\theta^\omega(\boldsymbol{x}_t^{\text{src}}, y^{\text{src}}, t)$ as Collaborative Score Distillation (CSD) (Kim et al., 2023), because contents in target and source prompts are often intersected. This setting prevents interruption in guidance toward the intersected contents. The image and text-guided noise prediction is calculated as follows:

$$\epsilon_\theta^\omega(\boldsymbol{x}_t^{\text{tgt}}, y^{\text{tgt}}, t) = \epsilon_\theta(\boldsymbol{x}_t^{\text{tgt}}, y_\varnothing, t) + \omega_y(\epsilon_\theta(\boldsymbol{x}_t^{\text{tgt}}, y^{\text{tgt}}, \tilde{\boldsymbol{x}}^{\text{src}}, t) - \epsilon_\theta(\boldsymbol{x}_t^{\text{tgt}}, y_\varnothing, \tilde{\boldsymbol{x}}^{\text{src}}, t))$$
$$+ \omega_I(\epsilon_\theta(\boldsymbol{x}_t^{\text{tgt}}, y_\varnothing, \tilde{\boldsymbol{x}}^{\text{src}}, t) - \epsilon_\theta(\boldsymbol{x}_t^{\text{tgt}}, y_\varnothing, \tilde{\boldsymbol{x}}_\varnothing, t)), \quad (17)$$
$$\epsilon_\theta^\omega(\boldsymbol{x}_t^{\text{src}}, y^{\text{src}}, t) = \epsilon_\theta(\boldsymbol{x}_t^{\text{src}}, y_\varnothing, t) + \omega_I(\epsilon_\theta(\boldsymbol{x}_t^{\text{src}}, y_\varnothing, \tilde{\boldsymbol{x}}^{\text{src}}, t) - \epsilon_\theta(\boldsymbol{x}_t^{\text{src}}, y_\varnothing, \tilde{\boldsymbol{x}}_\varnothing, t)), \quad (18)$$

where $\omega_I$ is a scale of image-guidance, and $\tilde{\boldsymbol{x}}^{\text{src}}$ and $\tilde{\boldsymbol{x}}_\varnothing$ are embeddings of a source image and a null-image, respectively.

## C ADDITIONAL QUALITATIVE EVALUATION

### C.1 COMPARISON WITH PDS USING DIFFUSION REVERSE PROCESS

As shown in Fig. 8, we compare our method with PDS when using an approximated diffusion reverse process with decreasing timestep scheduling in NeRF scenes. PDS, originally designed with random timestep sampling, performs poorly in this setting, leading to significant loss of fine details and identity features.

For instance, in the first row of Fig. 8, the model generates a highly distorted figure with blurred outlines and missing structural detail in PDS. However, our method preserves the subject's physical characteristics. The same behavior is observed in the second and third rows, where the objects generated by PDS appear more synthetic, lacking texture and natural form. Specifically, in the case of the third row, PDS loses the facial structure of the source image and produces an unrealistic outcomes.

This comparison underscores a limitation of applying the approximated diffusion reverse process with decreasing timestep scheduling in PDS. While this approach may offer faster edits, it comes at the expense of sacrificing essential identity features and fine-grained details. On the other hand, our method effectively leverages the diffusion reverse process while maintaining the identity of the source scene.

### C.2 COMPARISON IN NERF SCENES

We compare the results of our method the figures provided in PDS (Koo et al., 2023). 7 incorporate teaser results of Batman and tulip examples from PDS. Additionally, we include a comparison of editing a face into a skull using results obtained from the official PDS project page (https://posterior-distillation-sampling.github.io). As a results, DreamCatalyst demonstrates more realistic editing results while preserving the background details, outperforming the representative results of PDS.

### C.3 COMPARISON IN 3DGS SCENES

Fig. 9 presents additional qualitative results for methods specifically designed for 3DGS, including Gaussian Editor (GE) and DGE. Since our method is adaptable to various 3D editing frameworks

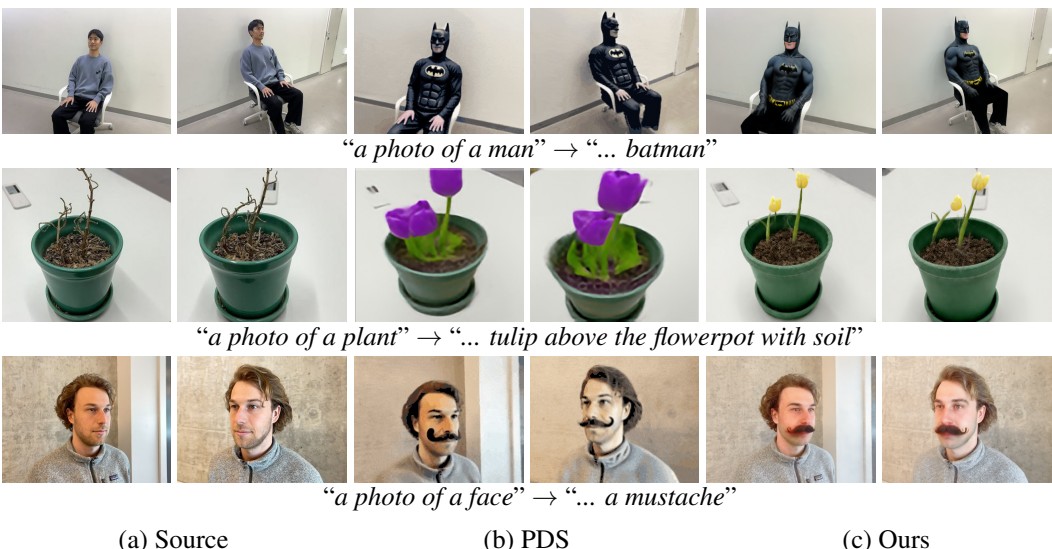

*"a photo of a man" → "... batman"*

*"a photo of a plant" → "... tulip above the flowerpot with soil"*

*"a photo of a face" → "... a mustache"*

| (a) Source | (b) PDS | (c) Ours |

**Figure 7: Qualitative comparison with the original PDS paper.** We compare our results with those presented in the original PDS paper. The top two rows show results from the teaser figure in the original PDS paper, and the last row is from a figure on the official project page of PDS (https://posterior-distillation-sampling.github.io).

through score distillation sampling, we also evaluate GE's performance when integrated with our approach.

Both GE and DGE rely on 2D segmentation masks for partial editing, which are then projected into 3D. However, unprojecting 2D segmentation masks onto 3D scenes often results in inconsistencies, as the masks may not accurately align with the 3D structure. Additionally, the segmentation text prompt may not fully correspond to the intended editing target, leading to unintended areas being included in the editing region. For instance, in the first and second rows of Fig. 9, GE introduces unintended edits to irrelevant regions based on the provided prompts, such as altering the clothing in the Einstein image or the arms of the clown. Moreover, in the third row, both baselines fail to generate the tulip on the plant as specified by the target prompt. In contrast, the Gaussian Editor combined with our method demonstrates a better trade-off between editing quality and identity preservation.

Further qualitative comparisons between model-agnostic methods in 3DGS scenes are shown in Fig. 10. We compare our method with 3D editing results from PDS without any refinement stage to ensure a fair comparison. In the first, second, and fourth rows of Fig. 10, PDS struggles to maintain the structural integrity of the source and generates blurry, over-saturated results. Additionally, in the third row, PDS fails to edit the man into a Hulk as required by the target prompt. In contrast, DreamCatalyst produces more accurate, detailed, and visually superior edited results.

## C.4 More Qualitative Ablation on FreeU

As seen in Fig. 11, different values of $b$ result in varying levels of editability and identity preservation. When $b = 1.0$, the model often struggles to fully align with the text prompts, leading to incomplete or subtle edits. For example, in the second row of Fig. 11, the editing with $b = 1.0$ fails to apply the snow effect across the scene. Similarly, in the fourth row, the model fails to render the elf-like green hair, suggesting that a low $b$ hinders the model's ability to achieve complete editing. Conversely, when $b$ is increased to $1.3$, over-editing becomes more pronounced, often resulting in unnatural outcomes or edits being applied to unintended areas. In the first row of Fig. 11, we observe that the Joker's face is reflected in the background wall, an unintended consequence of over-editing. Likewise, in the third row, the sunflower appears both in the pot and the background, further demonstrating that $b = 1.3$ leads to excessive and unrealistic editing. Our proposed setting, $b = 1.1$, achieves an optimal balance between identity preservation and editability. Thus, using FreeU with an adequate value of $b$ produces editing results that are visually coherent while aligning with the text prompt without introducing artifacts or over-editing as seen in higher values of $b$.

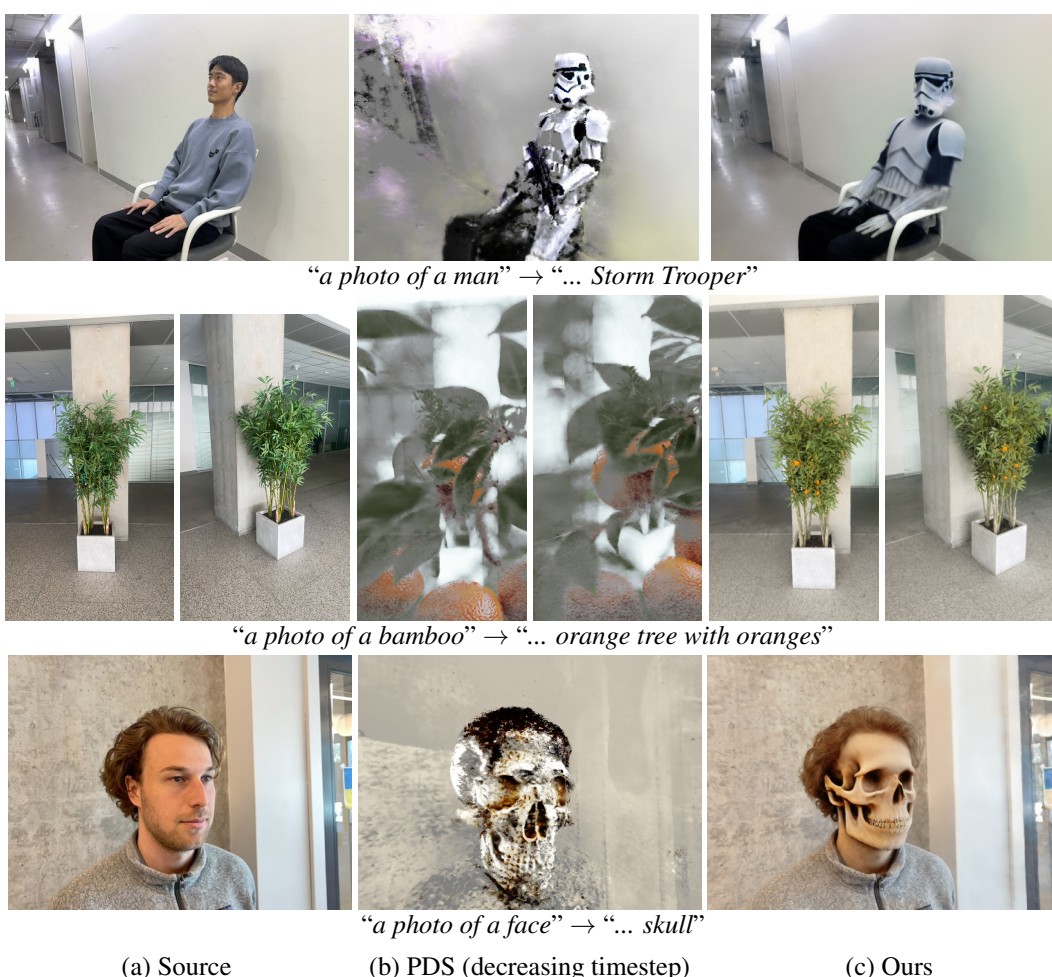

*"a photo of a man"* → *"... Storm Trooper"*

*"a photo of a bamboo"* → *"... orange tree with oranges"*

*"a photo of a face"* → *"... skull"*

(a) Source          (b) PDS (decreasing timestep)          (c) Ours

Figure 8: **Qualitative comparison with PDS using diffusion reverse process with decreasing timestep scheduling in NeRF scenes.** When applying the PDS method with decreasing timestep scheduling, the results lose fine details and fail to preserve key identity features.

Furthermore, we integrate FreeU with PDS to qualitatively evaluate the effect of FreeU. As shown in 13, PDS with FreeU tends to produce over-edited results and more background distortions compared to the original PDS. This indicates that the effect of FreeU is sensitive to the editing capabilities of the baseline method. In contrast, DreamCatalyst without FreeU already produces more realistic and visually appealing edits compared to PDS with FreeU. Thus, DreamCatalyst effectively balances editability and identity preservation by combining modified loss weighting and FreeU.

# D ADDITIONAL QUANTITATIVE EVALUATION

## D.1 COMPARISON IN CONVERGENCE SPEED

In this section, we highlight DreamCatalyst's faster convergence speed compared to other methods. 12 quantitatively demonstrates that DreamCatalyst converges significantly faster than the baseline methods. For this evaluation, we utilized CLIP Directional similarity as a metric to reflect the editing convergence behavior, since CLIP image similarity and Aesthetic score do not adequately capture the editing convergence behavior. 14 presents qualitative results highlighting the editing convergence. These results indicate that DreamCatalyst achieves substantially faster convergence compared to PDS and IN2N.

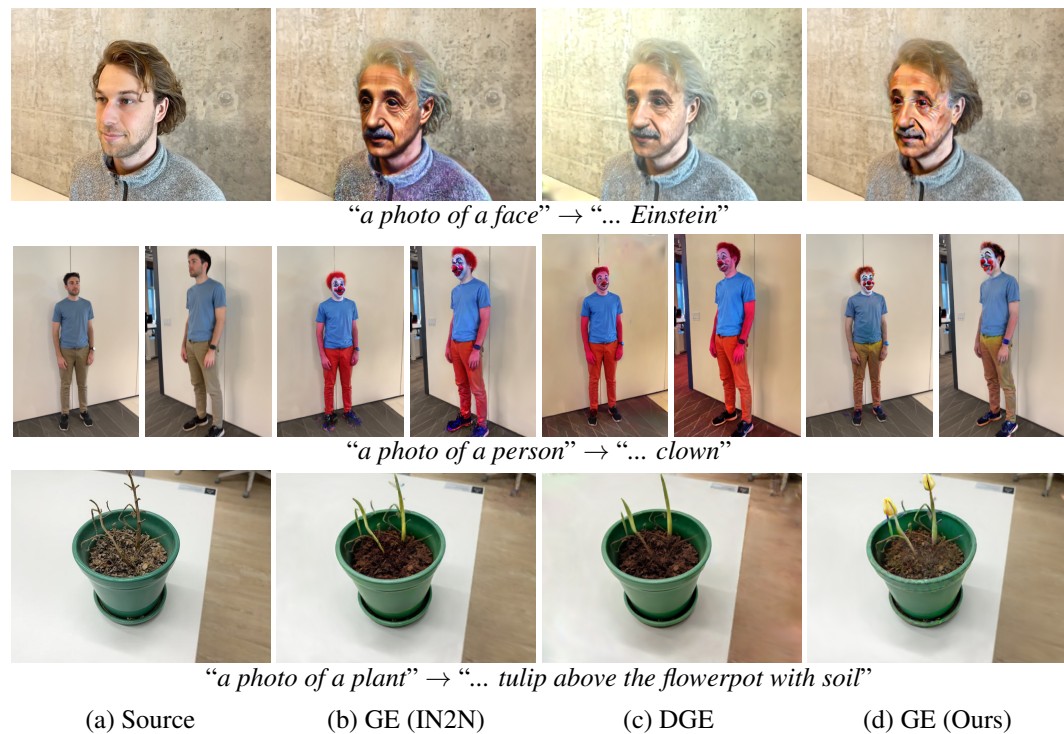

"a photo of a face" → "... Einstein"

"a photo of a person" → "... clown"

"a photo of a plant" → "... tulip above the flowerpot with soil"

(a) Source      (b) GE (IN2N)      (c) DGE      (d) GE (Ours)

Figure 9: **Qualitative comparison with model-specific baseline methods on 3DGS scenes.**

## D.2 QUANTITATIVE ABLATION ON FREEU

We conduct experiments applying FreeU to both PDS and DreamCatalyst, as in Table 5. We set the FreeU hyperparameter $b = 1.1$ for PDS, consistent with its configuration in DreamCatalyst. The results show a slight increase in the CLIP-Directional Similarity, from 0.161 to 0.162, indicated enhanced editability with the use of FreeU. However, both the CLIP Image Similarity and Aesthetic Score decreased—from 0.687 to 0.668 and from 5.437 to 5.413, respectively. This decline can be attributed to PDS underweighting identity preservation at large timesteps as in 2a, leading to insufficient preservation of identity features. The improved editability from FreeU exacerbates this issue, resulting in a loss of original identity and the generation of unrealistic image outputs, as detailed in Appendix C.4 and visualized in 13.

However, integrating FreeU into our method led to significant improvements in both the CLIP-Directional Similarity (from 0.171 to 0.180) and the Aesthetic Score (from 5.564 to 5.688), highlighting enhanced editability and visual quality. DreamCatalyst achieves an effective balance between editability and identity preservation by combining modified loss weighting with FreeU. These results indicate that while FreeU can enhance editability metrics, its effectiveness depends on the underlying method's ability to preserve identity features and produce realistic images. Therefore, combining our modified loss weighting with FreeU is essential for achieving superior results in DreamCatalyst.

Table 5: **Ablation on FreeU for PDS.** Quantitative ablation on the effects of FreeU for PDS and ours.

| Model | FreeU | CLIP-Direc (↑) | CLIP-Img (↑) | Aesthetic (↑) |
|-------|-------|----------------|--------------|---------------|
| PDS | ✗ | 0.161 | 0.687 | 5.437 |
| PDS | ✓ | 0.162 | 0.668 | 5.413 |
| Ours | ✗ | 0.171 | 0.744 | 5.564 |
| Ours | ✓ | 0.180 | 0.746 | 5.688 |

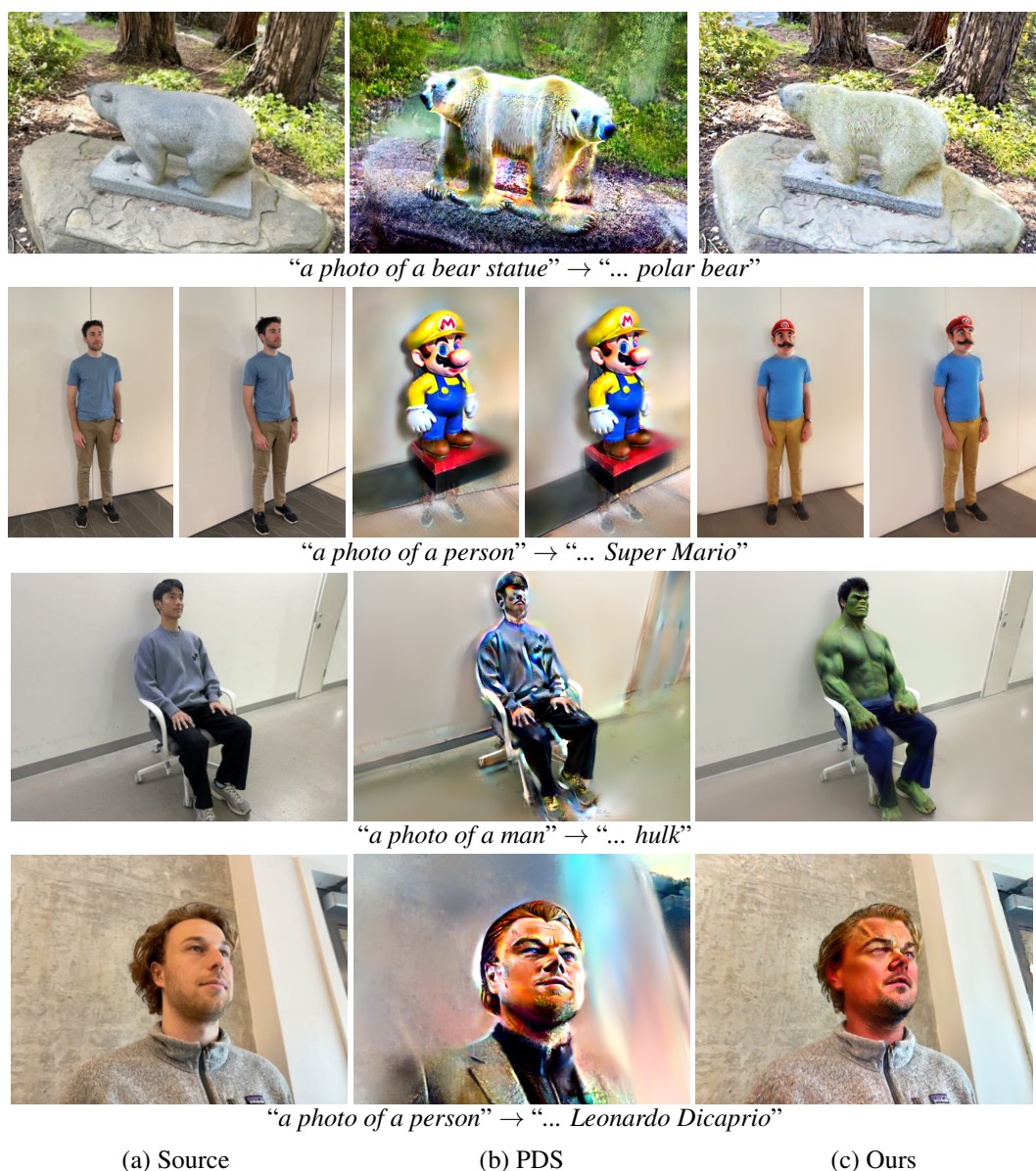

"*a photo of a bear statue*" → "*... polar bear*"

"*a photo of a person*" → "*... Super Mario*"

"*a photo of a man*" → "*... hulk*"

"*a photo of a person*" → "*... Leonardo Dicaprio*"

(a) Source         (b) PDS         (c) Ours

Figure 10: **Qualitative comparison with model-agnostic baseline methods on 3DGS scenes.**

# E  EXPERIMENTAL DETAILS

## E.1  EVALUATION METRICS

A qualitative assessment of DreamCatalyst was conducted using three evaluation methods: CLIP image similarity, CLIP directional similarity, and aesthetic score. The CLIP image similarity metric (Hessel et al., 2021) quantifies the visual similarity between original and edited images based on CLIP embeddings, ensuring visual consistency between them. CLIP directional similarity metric (Gal et al., 2022) quantifies the alignment of the changes between two text captions (ground-truth and edited) with the changes in two images (ground-truth and edited). Lastly, the aesthetic score metric quantifies the overall quality of the edited image, with the LAION Aesthetic Predictor (Schuhmann, 2022).

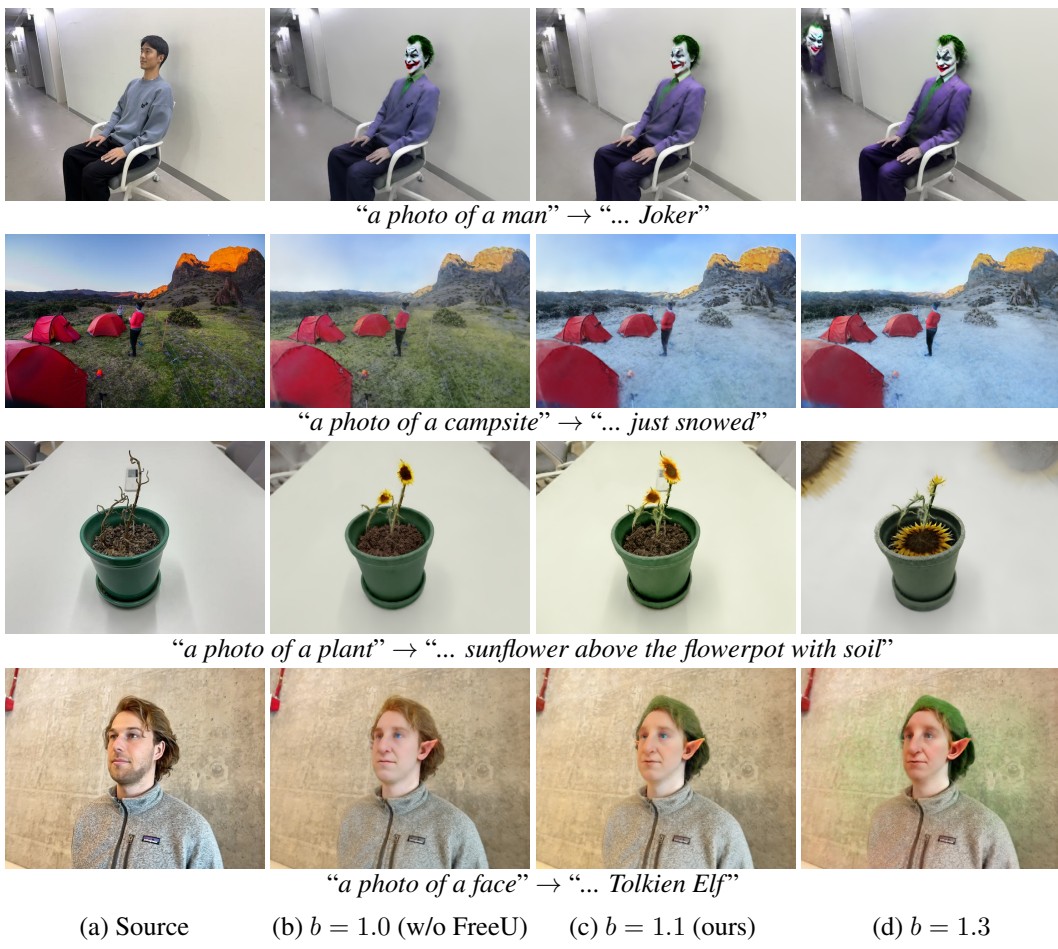

*"a photo of a man"* → *"... Joker"*

*"a photo of a campsite"* → *"... just snowed"*

*"a photo of a plant"* → *"... sunflower above the flowerpot with soil"*

*"a photo of a face"* → *"... Tolkien Elf"*

| (a) Source | (b) $b = 1.0$ (w/o FreeU) | (c) $b = 1.1$ (ours) | (d) $b = 1.3$ |

Figure 11: **More qualitative ablation on FreeU for our method in NeRF editing.** The figure illustrates the effects of varying $b$ values on editability and identity preservation. Lower $b$ values struggle to fully apply the edits, while higher $b$ values lead to over-editing.

### E.2    EVALUATION TEXT PROMPTS

Since IN2N (Haque et al., 2023), GE (Chen et al., 2024b), DGE (Chen et al., 2024a), and our method are based on IP2P (Brooks et al., 2023), these are specialized in processing instruction-style text prompts. In contrast, PDS utilizes the Stable Diffusion model (Rombach et al., 2022), which is designed to understand description-style text prompts. Therefore, we conduct experiments using pairs of corresponding description-style and instruction-style prompts curated from prior works (Haque et al., 2023; Koo et al., 2023) or generated by GPT-4 (OpenAI, 2023). We then conduct experiments on PDS using description-style text prompts (e.g., *"a photo of a Batman"*), while IN2N, GE, DGE, and our method utilize instruction-style prompts (e.g., *"Turn him into a Batman"*). The detailed list of text prompts used for the evaluation can be found in Tab. 6

### E.3    USER STUDY DETAILS

We recruited 100 participants using Amazon Mechanical Turk to ensure a fair comparison. To maintain consistent evaluations of the 3D scenes, we followed the same camera trajectory for each scene and recorded rendered videos of both the source and edited NeRF scenes. To ensure data quality, we included vigilance tasks, asking participants to identify trivially incorrect edited results (e.g., edited scenes from different source scenes) as in Fig. 15. Only participants who passed these tasks were included in the analysis.

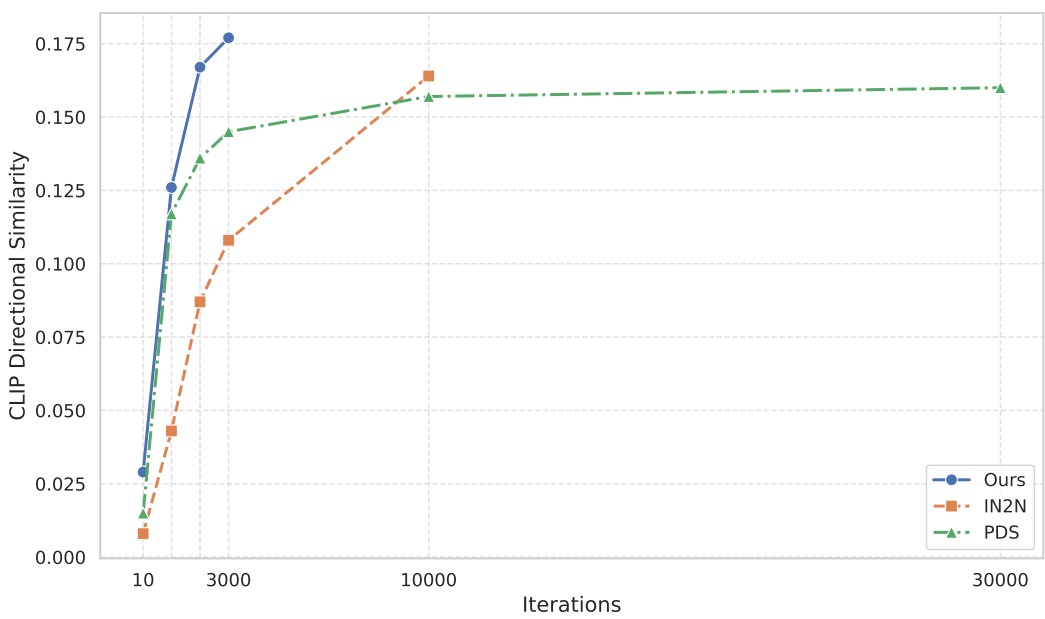

**Figure 12: Comparison of CLIP Directional Similarity across different iterations for ours, IN2N, and PDS models.**

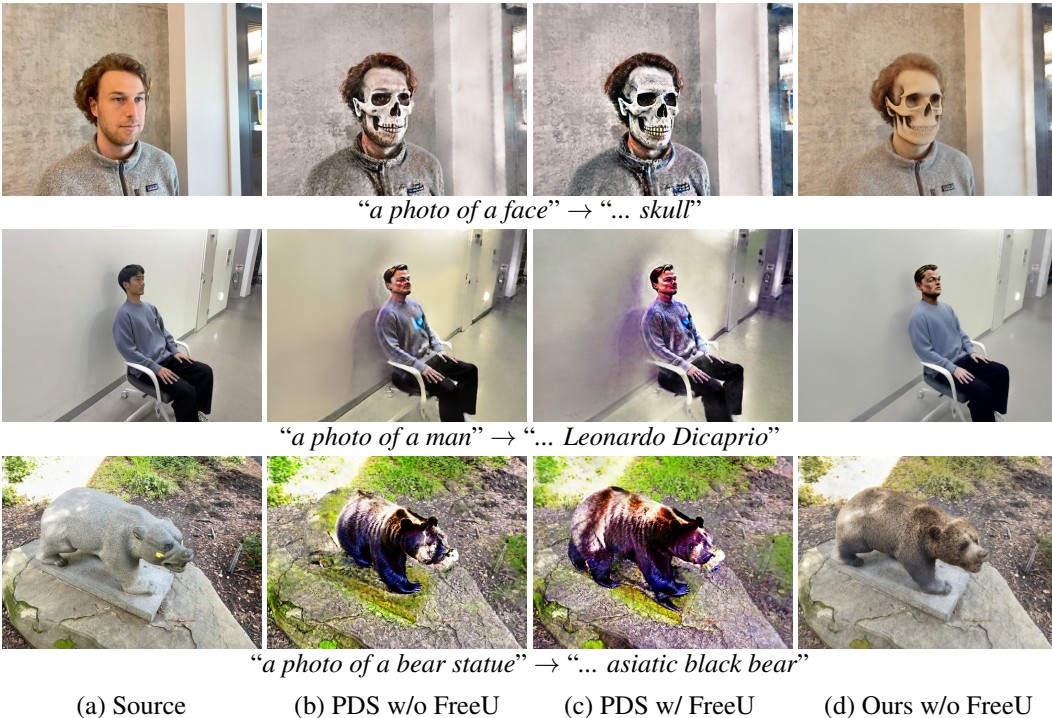

"*a photo of a face*" → "*... skull*"

"*a photo of a man*" → "*... Leonardo Dicaprio*"

"*a photo of a bear statue*" → "*... asiatic black bear*"

|  (a) Source | (b) PDS w/o FreeU | (c) PDS w/ FreeU | (d) Ours w/o FreeU |

**Figure 13: More qualitative ablation on FreeU for PDS.** We evaluate the effect of adding FreeU to PDS. Each column shows: (a) Source image, (b) Original PDS result (without FreeU), (c) PDS with FreeU, (d) Our method without FreeU.

Participants were shown a rendered video of a source scene along with the corresponding source prompt. They were then asked to evaluate the edited scenes generated by different baselines, including IN2N (Haque et al., 2023), PDS (Koo et al., 2023), and our proposed method as shown in

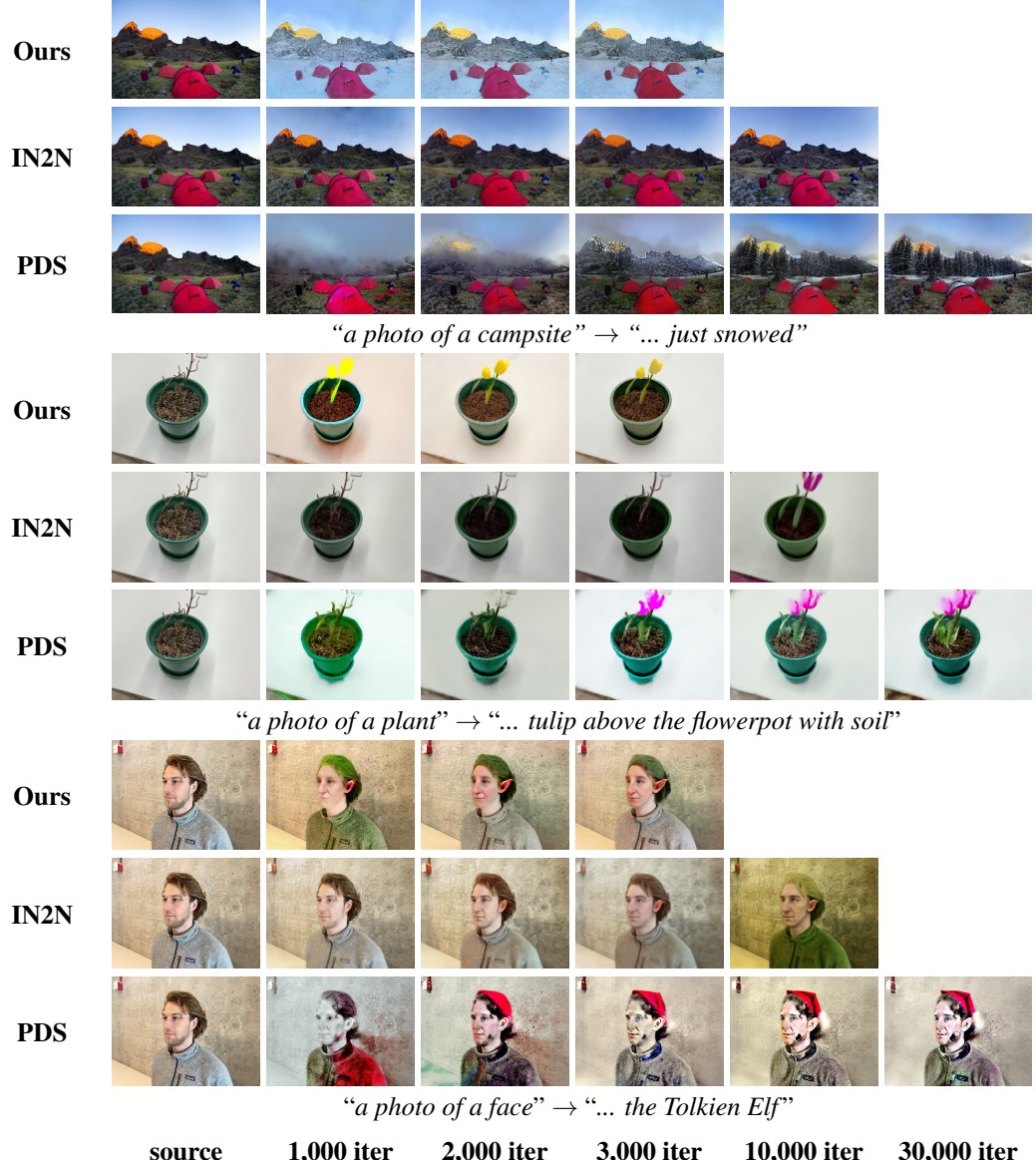

**Figure 14: Qualitative comparison of different models across editing iterations for different editing prompts.**

Fig. 16. Each baseline was edited based on a specific target prompt, and the baseline options were randomly shuffled for each evaluation.

Participants were instructed to select the best result for each criterion, based on the following:

1. **Prompt alignment (Editability):** When editing the video, which video best aligns with the text prompt?

2. **Identity preservation:** When editing the source video, which edited video best preserves the background and identity of the source video?

3. **Overall quality:** When editing the video, which video shows the best editing quality?

| Scene | Source Prompt | Target Prompt (Description) | Target Prompt (Instruction) |
|---|---|---|---|
| Face | "a photo of a face" | "a photo of the Tolkien Elf" | "Turn him into the Tolkien Elf" |
| Face | "a photo of a face" | "a photo of the Emma Watson" | "Turn him into Emma Watson" |
| Face | "a photo of a face" | "a photo of Elon Musk" | "Turn him into Elon Musk" |
| Face | "a photo of a face" | "a photo of an Einstein" | "Turn him into an Einstein" |
| Face | "a photo of a face" | "a photo of a face with mustache" | "Give him a mustache" |
| Face | "a photo of a face" | "a photo of Leonardo Dicaprio" | "Turn him into Leonardo Dicaprio" |
| Face | "a photo of a face" | "a photo of a skull" | "Turn his face into a skull" |
| Bear | "a photo of a bear statue" | "a photo of a grizzly bear" | "Turn the bear statue into a grizzly bear" |
| Bear | "a photo of a bear statue" | "a photo of a panda" | "Turn the bear statue into a panda" |
| Bear | "a photo of a bear statue" | "a photo of an asiatic black bear" | "Turn the bear statue into an asiatic black bear" |
| Bear | "a photo of a bear statue" | "a photo of a polar bear" | "Turn the bear statue into a polar bear" |
| Bear | "a photo of a bear statue" | "a photo of a Bengal tiger" | "Turn the bear statue into a Bengal tiger" |
| Bear | "a photo of a bear statue" | "a photo of a skull bear" | "Turn the bear statue into a skull bear" |
| Person | "a photo of a person" | "a photo of a Super Mario" | "Turn him into a Super Mario" |
| Person | "a photo of a person" | "a photo of a clown" | "Turn him into a clown" |
| Person | "a photo of a person" | "a photo of a person reading a book" | "Make him reading a book" |
| Person | "a photo of a person" | "a photo of a person wearing a sunglass" | "Wear him a sunglass" |
| Person | "a photo of a person" | "a photo of a Storm Trooper" | "Turn him into a Storm Trooper" |
| Person | "a photo of a person" | "a photo of Elon Musk" | "Turn him into Elon Musk" |
| Person | "a photo of a person" | "a photo of an Iron Man" | "Turn him into an Iron Man" |
| Person | "a photo of a person" | "a photo of a Jack Sparrow" | "Turn him into a Jack Sparrow" |
| Person | "a photo of a person" | "a photo of a bronze statue" | "Make him into a bronze statue" |
| Plant | "a photo of a plant" | "a photo of a rose above the flowerpot with soil" | "Turn only the plant above the flowerpot into a rose and keep soil" |
| Plant | "a photo of a plant" | "a photo of a sunflower above the flowerpot with soil" | "Turn only the plant above the flowerpot into a sunflower and keep soil" |
| Plant | "a photo of a plant" | "a photo of a tulip above the flowerpot with soil" | "Turn only the plant above the flowerpot into a tulip and keep soil" |
| Bamboo | "a photo of a bamboo" | "a photo of an orange tree with oranges" | "Turn the tree into an orange tree with oranges" |
| Campsite | "a photo of a campsite" | "a photo of a campsite just snowed" | "Make it look like just snowed" |
| Campsite | "a photo of a campsite" | "a photo of a campsite at sunset" | "Make it sunset" |
| Farm | "a photo of a farm" | "a photo of a farm in autumn" | "Make it autumn" |
| Yuseung | "a photo of a man" | "a photo of a Batman" | "Turn him into a Batman" |
| Yuseung | "a photo of a man" | "a photo of a Marvel's Spider-Man" | "Turn him into a Marvel's Spider-Man" |
| Yuseung | "a photo of a man" | "a photo of a clown" | "Turn him into a clown" |
| Yuseung | "a photo of a man" | "a photo of a Joker" | "Turn him into a Joker" |
| Yuseung | "a photo of a man" | "a photo of a Hulk" | "Turn him into a Hulk" |
| Yuseung | "a photo of a man" | "a photo of a Thanos" | "Turn him into a Thanos" |
| Yuseung | "a photo of a man" | "a photo of Leonardo Dicaprio" | "Turn him into Leonardo Dicaprio" |
| Yuseung | "a photo of a man" | "a photo of a Darth Vader" | "Turn him into a Darth Vader" |
| Yuseung | "a photo of a man" | "a photo of a Storm Trooper" | "Turn him into a Storm Trooper" |
| Yuseung | "a photo of a man" | "a photo of a Deadpool" | "Turn him into a Deadpool" |
| Yuseung | "a photo of a man" | "a photo of a baldman" | "Make him a bald" |

Table 6: **Detailed evaluation text prompts.** The description-style prompts are used for PDS, which employs the Stable Diffusion model (Rombach et al., 2022), while the instruction-style prompts are applied to IN2N, GE, DGE, and our method, all of which utilize the InstructPix2Pix model (Brooks et al., 2023).

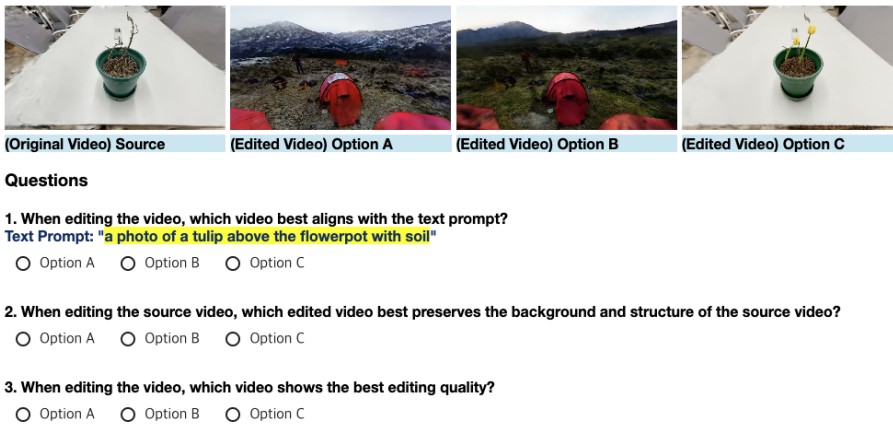

Figure 15: **Vigilance test for user study.** The vigilance test was used to ensure participant attentiveness during the user study. Participants were asked to identify trivially incorrect edited videos, such as those with irrelevant or mismatched prompts and edits.

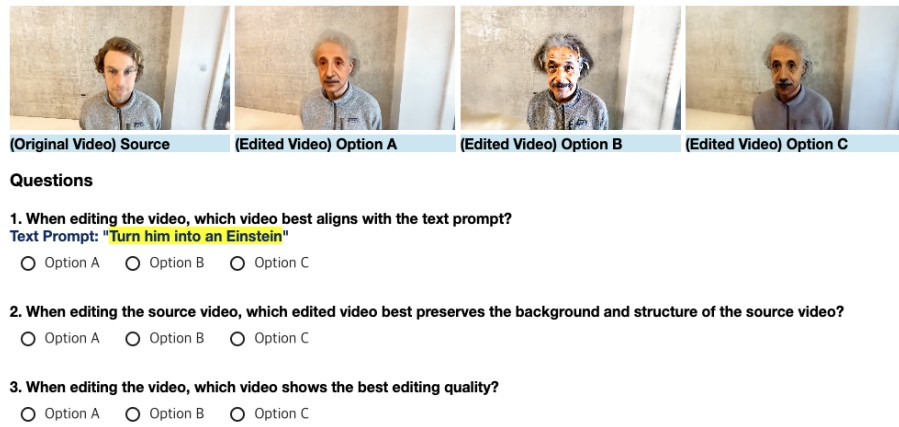

Figure 16: **User study.** Participants evaluated edited videos based on three criteria: prompt alignment, identity preservation, and overall quality. Each video was randomly shuffled for each evaluation to ensure unbiased selections. The target text prompts were highlighted in the prompt alignment questions.

# F    MORE VISUAL RESULTS

## F.1    RESULTS FROM DIFFRENT MODES IN DREAMCATALYST

In Fig. 17, we present additional visual results demonstrating the effectiveness of DreamCatalyst in both its fast and high-quality modes. Across a variety of scenes, DreamCatalyst preserves both editability and identity consistency. In the first row, for the target prompt of turning a campsite scene into a sunset, both modes produce convincing results with natural lighting, though minor color variations are visible between the fast and high-quality modes. Similarly, in the second row, when editing a bear statue into a panda, the fast mode accurately captures the desired editing, while the high-quality mode refines the textures for more photorealistic details. For more complex cases, such as in the third, fourth, and fifth rows, the high-quality mode captures finer details, while the fast mode still maintains overall structural accuracy. These examples highlight the ability of our method to achieve high visual fidelity while balancing computational efficiency across different modes, making it adaptable to varying time constraints and quality requirements.

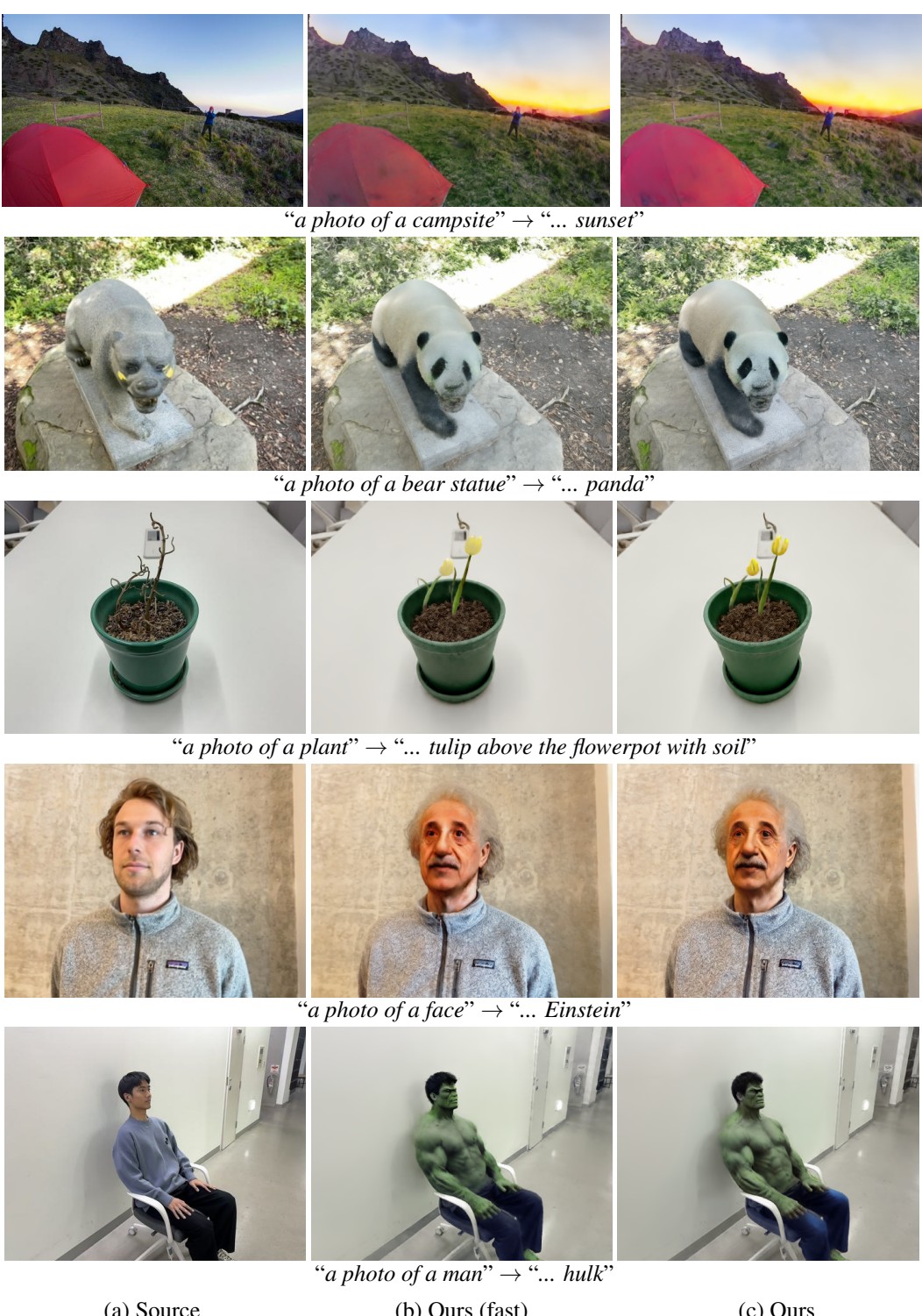

Figure 17: **Visual results from DreamCatalyst in different modes.** The fast mode of our method produces visually comparable results to the high-quality mode, while reducing editing time to approximately 35% of the high-quality mode.

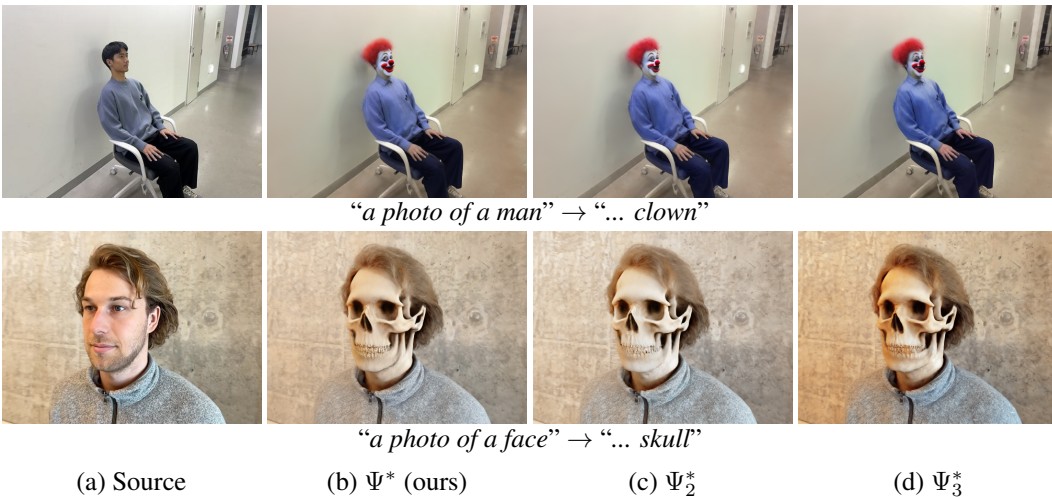

"*a photo of a man*" → "*... clown*"

"*a photo of a face*" → "*... skull*"

(a) Source      (b) $\Psi^*$ (ours)      (c) $\Psi_2^*$      (d) $\Psi_3^*$

Figure 18: **More visual results on weighting functions of $\mathcal{L}_{\text{DDS}}$ for NeRF editing.**

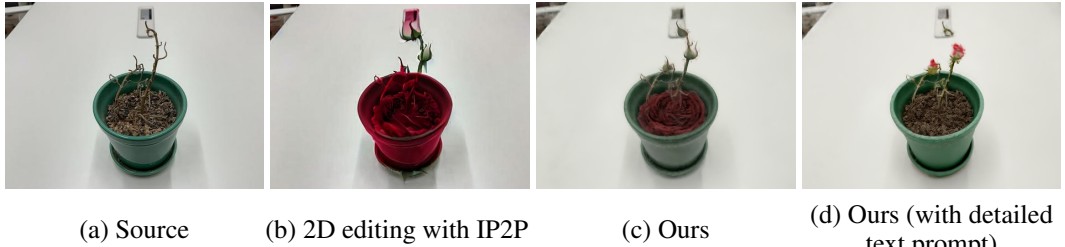

(a) Source    (b) 2D editing with IP2P    (c) Ours    (d) Ours (with detailed text prompt)

Figure 19: **Limitations.** The limitations of IP2P are reflected in the 3D editing process. In cases where IP2P struggles with 2D edits for a simple text prompt as in (b), this failure extends to the 3D editing as seen in our result with IN2N. The simple text prompt like "*Turn the plant into a rose*" fails to produce the desired outcomes as in (c). However, when using a more detailed prompt like "*Turn only the plant above the flowerpot into a rose and keep soil,*" our method successfully generates a rose on top of the branch as shown in (d).

### F.2    RESULTS FROM DIFFERENT WEIGHTING FUNCTIONS

As defined in Equation 14, $\Psi^*$ is a weighting function applied to $\mathcal{L}_{\text{DDS}}$ based on timesteps. As demonstrated in Fig. 2b, modifying the weighting function $\Psi^*$ to either $\Psi_2^*$ or $\Psi_3^*$ increases editability at lower timesteps. However, these variations do not significantly impact the results compared to $\Psi^*$. For instance, in both the first and second rows of Fig. 18, there is no major difference among the outputs from different weighting functions, leading to only subtle changes in the background.

Our findings confirm that fulfilling the two conditions mentioned in Sec. 3.1 enables effective 3D editing regardless of the specific choice of weighting function. However, we observe that inordinate editability in small timesteps rarely induces trivial color saturations on backgrounds—e.g., row 2 in 18. We hypothesize that the excessive editability during the final stages induces these color saturation artifacts. To prevent these color saturations, we design $\Psi^*(t)$ to drastically decrease editability in small timesteps. Thus, we utilize $\Psi^*$ as the default weighting function for $\mathcal{L}_{\text{DDS}}$, as supported by the quantitative results in Tab. 1. We leave the exploration of more optimal design choices for future work.

## G    ADDITIONAL DISCUSSIONS

### G.1    LIMITATIONS

DreamCatalyst uses IP2P (Brooks et al., 2023) as a pretrained diffusion model for distilling 2D prior knowledge in the 3D editing process. However, the limitations of IP2P can carry over into 3D

editing tasks. As shown in Fig. 19, when IP2P struggles with specific 2D edits, these limitations extend to our 3D editing results. If IP2P does not effectively edit certain prompts in 2D, the 3D editing may also fail to align with target text prompt. As IP2P or other 2D diffusion models improve in accuracy and prompt alignment, 3D editing capabilities of our method are expected to similarly improve, offering more reliable and precise outputs.

