# OpenReview forum: "DreamCatalyst: Fast and High-Quality 3D Editing via Controlling Editability and Identity Preservation"
_ICLR.cc/2025/Conference — ICLR 2025 Poster_

### Official Review · Reviewer_p9uk · 2024-11-02

**Soundness:** 3
**Presentation:** 3
**Contribution:** 3
**Rating:** 6
**Confidence:** 3

**Summary:**

The paper presents an innovative way to achieve 3D-consistent image editing on a 3D representation (NeRF/3DGS). It builds upon several foundational works (SDS, DDS, and PDS) to generate the edited scene with better quality and speed simultaneously. Novelty of the paper, written in section 3.3, is a (diffusion) time-step dependent weighting of the two primary loss terms (equation 17).

**Strengths:**

*The strength of this paper is the evaluation - the quantitative and qualitative evaluation includes the most recent works and demonstrate this method is the top-performer for this task. Table 1 shows that this paper’s family of models is faster and more semantically-aligned than existing work. Fig. 5 shows that this method generally achieves more favorable CLIP scores while being more efficient. Both NeRFs and 3DGS are evaluated.

**Weaknesses:**

*Technical novelty may be a bit limited. FreeU makes all Stable Diffusion models better. The core contribution appears to be a smart scheme to dynamically balance weights of an existing loss function.

**Questions:**

*4D editing of scenes is an active area of interest - could the authors comment on if/how this work could be adapted to such use-cases?

*Could you clarify what is novel about lines 349-350 r.e. “we adopt decreasing timestep sampling.” Isn’t this standard diffusion sampling (more noise to less noise)?

*Could the authors elaborate more on where exactly the time savings are achieved? Is it because fewer optimization steps are required to train the NeRF/3DGS after the author’s proposed modified loss/FreeU?

*Elaborate more on how equation 18 was obtained?

*Fig. 6 shows a qualitative ablation on FreeU. Could a quantitative evaluation be presented as well? How much of the improvements are due to FreeU vs. better loss weighting?

---

> ### Author Response · Authors · 2024-11-19
> **Response to Reviewer p9uk (Part 1/2)**
>
> Dear Reviewer p9uk,
>
> We thank the reviewer for their instructive feedback and thoughtful comments. Below, we provide detailed responses to each of the raised questions and concerns. Please let us know if further clarification is needed at any point.
>
> ---
>
> > **[W1, Q5] About FreeU**
>
> We remark that **our loss design is technically practical and efficient**. We would like to clarify that in Table 4, $b=1.0$ corresponds to DreamCatalyst without FreeU. Thus, Table 4 shows the quantitative ablation of FreeU. Notably, **DreamCatalyst without FreeU already achieves state-of-the-art results** compared to PDS and IN2N, as shown in Tables 1 and 4. This signifies our theoretically designed loss enables improved editing in both speed and quality. Specifically, **in fast mode, DreamCatalyst operates approximately 5 to 23 times faster than other baselines**, making 3D editing techniques more applicable in real-world scenarios by significantly reducing computation time.
>
> Furthermore, **DreamCatalyst introduces technical novelty by discovering FreeU’s strength in editing tasks and SDS.** To the best of our knowledge, we first discovered that FreeU is suitable for editing tasks without sacrificing identity preservation, not just for generation tasks. Also, instead of utilizing Dreambooth as PDS, we first employ FreeU to remove extra training time and computation time at the inference stage for the module in SDS and editing tasks. Moreover, the proposed formulation and FreeU have strong synergy. As shown in Table 6 and Figure 19, PDS with FreeU shows inferior results than the original PDS. This is because PDS underweights identity preservation at large timesteps, as shown in Figure 2 (a), leading to insufficient preservation of identity features. Enhancing editability with FreeU further exacerbates this issue, resulting in loss of the original identity and unrealistic image results, such as over-editing and background distortions as illustrated in Figure 19. These observations underscore the synergy between FreeU and our weight design rule, highlighting the technical novelty of our approach.
>
> We stress that **the performance improvements of our loss and FreeU are almost similar.** A comparison between Tables 1 and 4 reveals that the performance gains in key metrics—namely, the CLIP-Directional Similarity Score and Aesthetic Score—resulting from the modified loss weighting and FreeU are similar. This highlights the analogous importance of these two components in enhancing the editing quality within our framework. For your convenience, we provide integrated quantitative comparisons in Tables 1 and 4 as follows:
>
> | Method            | CLIP-Direc \($\uparrow$\) | CLIP-Img \($\uparrow$\) | Aesthetic \($\uparrow$\) |
> |---------------------|---------------------------|--------------------------|---------------------------|
> | DreamCatalyst                 | **0.180**                    | **0.746**                   | **5.688**                |
> | DreamCatalyst \(w/o FreeU\)     | *0.171*                    | *0.744*                   | *5.564*                    |
> | PDS                 | 0.161                | 0.687                   | 5.437                    |
> | IN2N                 | 0.157                | 0.722                   | 5.399                    |
>
> Note that **Bold** indicates the best result and *Italic* is the second-best result.
>
> ---
>
> > **[Q1] How can DreamCatalyst be adapted to 4D editing?**
>
> Extending 3D to 4D has recently garnered significant attention such as Monst3r, which extends Dust3r for 4D reconstruction. The key challenge in extending 3D to 4D lies in temporal understanding. However, standard 2D pretrained diffusion models do not contain any temporal priors and inherently incorporate only 3D priors. To enable 4D editing, DreamCatalyst must leverage pretrained video diffusion models, i.e., CogVideoX \[1\]. Such pretrained video diffusion models incorporate both 3D and temporal priors. By incorporating an **extra-temporal regularizer** into DreamCatalyst, 4D editing becomes feasible. As described in equation 16 (equation 13 in the revised manuscript), DreamCatalyst formulates SDS editing as an optimization problem, allowing for the inclusion of various regularizers, such as temporal regularization terms, to achieve 4D editing. **We believe that the suggested theoretical framework in DreamCatalyst can be generalized to a variety of tasks by appropriately adjusting the regularization terms.**
>
> ---
>
> **Reference**
>
> \[1\] Yang, Zhuoyi, et al. "Cogvideox: Text-to-video diffusion models with an expert transformer." arXiv preprint arXiv:2408.06072 (2024).

---

> ### Author Response · Authors · 2024-11-19
> **Response to Reviewer p9uk (Part 2/2)**
>
> > **[Q2] What is decreasing timestep sampling?**
>
> **We adopted decreasing timestep sampling to satisfy the equation 15 (equation 12 in the revised manuscript) at every timestep $t$.** For clarification on decreasing timestep sampling, please refer to common response 3. As we discussed in common response 3, SDS differs from standard diffusion sampling in its approach to iterative timestep sampling because SDS aims to distill the diffusion networks to the 3D model. To approximate the standard diffusion sampling, DreamCatalyst and DreamTime \[1\] sample $t$ consecutively (more noise to less noise) with SDS. The main difference between the decreasing timestep sampling and DreamTime lies in the sampling rate for each timestep $t$. By uniform decreasing timestep sampling, DreamCatalyst is more consistent with the standard diffusion process compared to DreamTime by almost satisfying the equation 15 at every $t$, as discussed in common response 3.
>
> ---
>
> > **[Q3] Why is DreamCatalyst fast?**
>
> In common response 1, we clarify why DreamCatalyst achieves faster 3D editing. Please refer to common response 1. **In short, our modified loss significantly reduces the required optimization steps, while FreeU further decreases editing time by removing the additional computational overhead associated with extra networks**, as seen in methods like DreamBooth and LoRA. Moreover, we carefully revised the manuscript to incorporate your thoughtful suggestions (lines 341-348 and 376-377).
>
> ---
>
> > **[Q4] Designing $\Phi$ and $\Psi$**
>
> We elaborated on why DreamCatalyst can save the editing time in common response 2. Please refer to common response 2.
>
> ---
>
> **Reference**
>
> \[1\] Huang, Yukun, et al. "DreamTime: An Improved Optimization Strategy for Text-to-3D Content Creation." *arXiv preprint arXiv:2306.12422* (2023).

---

> ### Author Response · Authors · 2024-11-23
> **Reminder**
>
> Dear Reviewer p9uk,
>
> Thank you for reviewing our paper and for providing valuable feedback. We’d like to check if you have any further concerns or comments that we can address. Please let us know if there’s anything else you’d like us to clarify or improve.
>
> Best regards,
>
> Authors

---

> ### Author Response · Authors · 2024-11-25
> **Sincerely looking forward to more discussion with you**
>
> Dear Reviewer p9uk,
>
> The discussion phase has only two days remaining, and we thus kindly request you to let us know if our response has addressed your concerns. If there are additional issues or questions, we would be happy to address them. Otherwise, we would greatly appreciate it if you could consider updating your score to reflect that the issues have been resolved.
>
> Best regards,
>
> Authors

---

> ### Author Response · Authors · 2024-11-26
> **Last day for revising the manuscript**
>
> Dear Reviewer p9uk,
>
> We sincerely appreciate your thoughtful suggestions. As your helpful feedback, we have revised the manuscript and highlighted the changes in red as follows:
>
> - [Q3] We have clarified how DreamCatalyst achieves fast editing in the revised manuscript (lines 341-348 and 376-377).
> - [Q5] In Table 4, $b=1.0$ corresponds to DreamCatalyst without FreeU, illustrating the quantitative ablation on FreeU. We have updated the notation to explicitly label $b=1.0$ as "DreamCatalyst w/o FreeU" in the revised manuscript.
>
> As today is the last day for submitting revisions (with six days remaining in the discussion period), we wanted to inform you that the revision phase is concluding. If you have any additional suggestions or concerns, please let us know at your earliest convenience. We are eager to discuss and address any further feedback you may have during the remaining discussion period.
>
> Best regards,
>
> Authors

---

> ### Author Response · Authors · 2024-12-01
> **Only two days left in the discussion period**
>
> Dear Reviewer p9uk,
>
> We hope this message finds you well.
> As the discussion period is nearing its conclusion in just two days, we wanted to check if we have sufficiently addressed your concerns and questions.
> We would greatly appreciate any further discussion to address your concerns. Looking forward to hearing your thoughts!
>
> Best regards,
>
> Authors

---

> ### Author Response · Authors · 2024-12-02
> **Only 24 hours left in the discussion period**
>
> Dear Reviewer p9uk,
>
> As there are only 24 hours left in the discussion period, we wanted to check if we have adequately addressed your concerns. We would greatly appreciate it if you could share your thoughts and engage in further discussion regarding our responses. Once again, thank you for your valuable time in reviewing our manuscript.
>
> Best regards,
>
> Authors

---

> ### Author Response · Authors · 2024-12-03
> **[Reminder] 8 hours left in the discussion period**
>
> Dear Reviewer p9uk,
>
> With only 8 hours remaining in the discussion period, we wanted to check if we have adequately addressed your concerns. Once again, thank you for dedicating your valuable time to reviewing our manuscript.
>
> Best regards,
>
> Authors

---

### Official Review · Reviewer_Hf3Z · 2024-11-03

**Soundness:** 3
**Presentation:** 3
**Contribution:** 2
**Rating:** 5
**Confidence:** 4

**Summary:**

In this paper, the author proposes DreamCatalyst, a method for editing 3D scene using improved Posterior Distillation Sampling loss. Based on the analysis of PDS loss, the authors proved that the coefficients of ID-preserving loss and the DDS loss can be independently selected under DDIM inversion. They also proposed several rules for setting these coefficients under different time steps. As a result of these advances, DreamCatalyst out-performs previous 3D editing methods in both speed and quality.

**Strengths:**

1. This paper is well-written and easy to follow.
2. The analysis of PDS loss is interesting.
3. Experiments show that the proposed method achieves good 3D editing results with faster speed.

**Weaknesses:**

1. The method proposed in the paper is actually just a supplement to PDS. The theoretical analysis merely shows that the weights of the two losses can be adjusted, a fact that was already discovered and utilized in previous methods like Fantasia3D and ProlificDreamer.
2. Some of the cases used in the experiments are already present in the original PDS paper. The results from the original paper should be used for these cases. However, the PDS results provided by the authors show a significant discrepancy from the original paper. I suggest that the authors compare their results with those in the original PDS paper. I will adjust my review based on these comparisons.

**Questions:**

1. Why does DreamCatalyst rely that heavily on FreeU? In fig.6, with $b=1$, the model performs poorly compared with the teaser figure of the PDS paper. Is there a reasonable explanation for this?
2. Just curious, is the determination of the functional forms of $\Psi$ and $\Phi$ based on better theoretical analysis or qualitative constraints? If it's qualitative analysis, what impact do other function families or parameters that meet the proposed conditions have on the results?

---

> ### Author Response · Authors · 2024-11-19
> **Response to Reviewer Hf3Z (Part 1/2)**
>
> Dear Reviewer Hf3Z,
>
> We greatly appreciate your insightful feedback and thoughtful comments. Below, we have provided detailed responses to each of your questions and concerns. Please do not hesitate to let us know if there are any points that need further explanation or additional clarification.
>
> ---
>
> > **[W1] Comparison to PDS**
>
> We stress that **our DreamCatalyst is a generalized formulation of PDS, positioning PDS as a special case of DreamCatalyst**, just as SDS is a special case of VSD \[1\]. Our theoretical analysis shows that PDS cannot change the weighting of loss terms and the regularization function itself (L2 loss in PDS). However, our analysis enables not only reweighting the loss terms but also the use of regularization functions. Our theory allows using other identity preservation loss terms. Thus, our theoretical analysis gives broader ways for future work such as extending DreamCatalyst to 4D editing, discussed in reviewer p9uk’s Q1.
>
> In addition, the main difference between DreamCatalyst and previous methods (i.e., ProlificDreamer and Fantasia3D \[2\]) lies in the purpose of utilizing multiple loss terms. (1) Fantasia3D disentangles geometry and appearance by separating them into independent models, each optimized with its own SDS loss. Unlike our approach, Fantasia3D uses the SDS loss independently for optimizing separate geometry and appearance models, without regularizing the SDS loss for a single model. (2) ProlificDreamer employs two loss terms: the SDS loss for finetuning the LoRA model and a separate loss term for optimizing the 3D model. Similar to Fantasia3D, each loss term in ProlificDreamer is dedicated to optimizing a distinct model. In contrast, DreamCatalyst leverages two loss terms exclusively to optimize a single 3D model. Specifically, **DreamCatalyst regularizes the SDS loss to enhance optimization for the single model, whereas ProlificDreamer and Fantasia3D employ multiple loss terms to independently optimize separate models, without regularizing for a unified objective.** These regularizations can be extended to future work of 3D editing (i.e., varying the identity preservation regularizer to improve editing results) and various tasks (e.g., 4D editing with additional temporal regularizer). Thus, our theoretical analysis has strong expandability for practical usage.
>
> ---
>
> > **[W2] The results of PDS**
>
> First, we fully followed the instructions of the official code of PDS. We observed that 3D editing results of PDS often show differences with their teaser figure as their GitHub issue \[3\]. However, we carefully revised the manuscript to provide qualitative comparisons with their teaser figure as your thoughtful suggestions to address your concern. **We brought the figures provided in the original PDS paper to compare with their best results.** In the revised manuscript, Figure 16 has been updated to incorporate teaser results of Batman and tulip examples from the original PDS paper (please refer to the revised manuscript’s Appendix). Additionally, we have included a comparison of editing a face into a skull using results obtained from the official PDS project page \[4\]. While the teaser primarily highlights two scenes, this additional example provides coverage of a different scene to further demonstrate the method’s versatility. As a result, DreamCatalyst demonstrates more realistic editing results while preserving the background details, outperforming the representative results of PDS. We sincerely appreciate your valuable feedback for the better manuscript.
>
> ---
>
> **Reference**
>
> \[1\] Wang, Zhengyi, et al. "Prolificdreamer: High-fidelity and diverse text-to-3d generation with variational score distillation." Advances in Neural Information Processing Systems 36 (2024).
>
> \[2\] Chen, Rui, et al. "Fantasia3d: Disentangling geometry and appearance for high-quality text-to-3d content creation." Proceedings of the IEEE/CVF international conference on computer vision. 2023.
>
> \[3\] https://github.com/KAIST-Visual-AI-Group/PDS/issues/7
>
> \[4\] https://posterior-distillation-sampling.github.io

---

> ### Author Response · Authors · 2024-11-19
> **Response to Reviewer Hf3Z (Part 2/2)**
>
> > **[Q1] Significance of FreeU**
>
> **DreamCatalyst without FreeU already achieves state-of-the-art results.** We clarify that $b=1.0$ in Table 4 indicates the configuration without FreeU and we have revised the manuscript to explicitly highlight it. Notably, the quantitative results without FreeU outperform the existing methods, as illustrated in Table 1. For your convenience, we provide integrated quantitative comparisons in Tables 1 and 4 as follows:
>
> | Method            | CLIP-Direc \($\uparrow$\) | CLIP-Img \($\uparrow$\) | Aesthetic \($\uparrow$\) |
> |---------------------|---------------------------|--------------------------|---------------------------|
> | DreamCatalyst                 | **0.180**                    | **0.746**                   | **5.688**                |
> | DreamCatalyst \(w/o FreeU\)     | *0.171*                    | *0.744*                   | *5.564*                    |
> | PDS                 | 0.161                | 0.687                   | 5.437                    |
> | IN2N                 | 0.157                | 0.722                   | 5.399                    |
>
> Note that **Bold** indicates the best result and *Italic* is the second-best result.
>
> **These quantitative results provide overall quality comparisons of PDS and DreamCatalyst without FreeU, while the Batman result in Figure 6 represents only a single case.** Furthermore, as in Figure 19 in the revised manuscript’s Appendix, qualitative comparisons reveal that DreamCatalyst without FreeU produces more realistic and visually appealing edits compared to PDS with FreeU. Therefore, the quantitative results demonstrate that DreamCatalyst does not heavily rely on FreeU.
>
> ---
>
> > **[Q2] Designing $\Phi$ and $\Psi$**
>
> In common response 2, we have discussed the determination of our formulation. Please refer to common response 2. In short, we determined the $\Phi$ and $\Psi$ with the proposed two conditions and our observation (discussed in common response 2). Furthermore, our design rule demonstrates robustness, as various function families following this rule produce consistent and reliable results.

---

> > ### Author Response · Authors · 2024-11-23
> > **Reminder**
> >
> > Dear Reviewer Hf3Z,
> >
> > Thank you once again for the time and effort you have dedicated to reviewing our paper. We greatly value your constructive feedback, which has been instrumental in enhancing the quality of our work. We would like to kindly inquire if there are any additional concerns or suggestions that we might address to further improve our submission.
> >
> > Best regards,
> >
> > Authors

---

> ### Author Response · Authors · 2024-11-25
> **Sincerely looking forward to more discussion with you**
>
> Dear Reviewer Hf3Z,
>
> The discussion phase has only two days remaining, and we thus kindly request you to let us know if our response has addressed your concerns. If there are additional issues or questions, we would be happy to address them. Otherwise, we would greatly appreciate it if you could consider updating your score to reflect that the issues have been resolved.
>
> Best regards,
>
> Authors

---

> ### Author Response · Authors · 2024-11-26
> **Last day for revising the manuscript**
>
> Dear Reviewer Hf3Z,
>
> We sincerely appreciate your helpful suggestions.  In response to your feedback, we have revised the manuscript and highlighted the changes in "red" as follows:
>
> - [W2] We have updated the qualitative comparisons between the teaser figures of PDS and DreamCatalyst in the revised manuscript (please refer to Figure 16).
>
> As today is the final day for revisions (with six days remaining in the discussion period), we wanted to inform you that the revision period is concluding. If you have any further suggestions or concerns, please let us know at your earliest convenience. We are eager to discuss and address any additional feedback you may have during the remaining discussion period.
>
> Best regards,
>
> Authors

---

> ### Author Response · Authors · 2024-12-01
> **Only two days left in the discussion period**
>
> Dear Reviewer Hf3Z,
>
> We hope this message finds you well.
> As the discussion period is nearing its conclusion in just two days, we wanted to check if we have sufficiently addressed your concerns, particularly regarding the comparison between DreamCatalyst and the best results of PDS.
> We would greatly appreciate any additional discussion to address your concerns. Looking forward to hearing your thoughts!
>
> Best regards,
>
> Authors

---

> ### Author Response · Authors · 2024-12-02
> **Only 24 hours left in the discussion period**
>
> Dear Reviewer Hf3Z,
>
> As there are only 24 hours left in the discussion period, we wanted to check if we have adequately addressed your concerns. We would greatly appreciate it if you could share your thoughts and engage in further discussion regarding our responses. Once again, thank you for your valuable time in reviewing our manuscript.
>
> Best regards,
>
> Authors

---

> ### Author Response · Authors · 2024-12-03
> **[Reminder] 8 hours left in the discussion period**
>
> Dear Reviewer Hf3Z,
>
> With only 8 hours remaining in the discussion period, we wanted to check if we have adequately addressed your concerns. Once again, thank you for dedicating your valuable time to reviewing our manuscript.
>
> Best regards,
>
> Authors

---

### Official Review · Reviewer_wbDz · 2024-11-04

**Soundness:** 3
**Presentation:** 2
**Contribution:** 3
**Rating:** 6
**Confidence:** 3

**Summary:**

This work presents DreamCatalyst, a variation of score distillation loss for the purpose of editing 3D scenes. This variation on SDS contains two terms: one based on DDS that controls the editing capabilities of the loss and one that is a regularization term intended to preserve the identity of the scene. The formulation in DreamCatalyst produces better quality edits and reduces edit time as compared to existing methods. The method is evaluated both qualitatively through many figures and quantitatively showing automated metrics as well as a perceptual user study.

**Strengths:**

Strengths:
- This work has promising results as the method shows impressive ability to edit only the regions indicated with the text prompt.
- This work proposes a reinterpretation of PDS loss that better aligns to the diffusion reverse sampling process.
- The proposed approach improves over existing techniques in both speed and edit quality.
- This work applies FreeU to the optimization to get better quality edits without sacrificing identity preservation.

**Weaknesses:**

Weaknesses:
- Given the similarity of DreamCatalyst to PDS, this work could benefit from a more clear / detailed discussion of the differences between these two approaches. Specifically, since the PDS loss in eq 14 in the PDS paper seems the same as eq 16 in this paper, my understanding is the main difference between these two is the hyperparameters that control the timestep dependent coefficients phi and psi for identity preservation and editability respectively. If this is the main difference, then it should be made more clear.
- Since the increased speed is a key contribution of this work, more space should be devoted to explaining how this approach actually does so, as it is not clear to me in the current state. It seems to be due to the timestep sampling and approximated diffusion reverse process. However, exactly why it is faster was not clear. Additionally, a helpful experiment to highlight the speed would be to show DreamCatalyst vs IN2N VS PDS on 1k, 3k, 15k, and 30k iterations so that we can see what the quality looks like for these other methods when DreamCatalyst converges.
- FreeU seems like an important component to increasing edit quality, but currently there don’t seem to be any experiments showing how important it is. While there is an ablation for the FreeU hyperparameter, an experiment comparing PDS and DreamCatalyst both with and without FreeU to see how much of an impact it makes would be helpful.

**Questions:**

Why is this method able to work with the approximated diffusion reverse process while standard PDS is not (Fig. 7). Is it just due to the coefficients phi and psi and in the case of PDS, these coefficients don't allow sufficient editability at small timesteps whereas DreamCatalyst’s do?

Minor questions:
- L:349-350 – “uniformly samples timestep t = T → 1” My understanding is that $t$ starts at $T$, ends at $1$, and then at an arbitrary iteration $i$, $t = T - i$. If this is the case, why is this uniform sampling? Maybe I am missing something here.

---

> ### Author Response · Authors · 2024-11-19
> **Response to Reviewer wbDz (Part 1/2)**
>
> Dear Reviewer wbDz,
>
> We sincerely appreciate your valuable feedback and constructive comments. Below, we address each concern and question in detail. Please let us know if there are any remaining issues or concerns that require further clarification. We carefully revised the manuscript to reflect the suggested qualitative and quantitative results in the Appendix.
>
> ---
>
> > **[W1] Comparison to PDS**
>
> We sincerely appreciate your thoughtful suggestions for improving the manuscript. The key distinction between DreamCatalyst and PDS lies in the design of $\Phi(t)$ and $\Psi(t)$. We remark that **PDS cannot modify $\Phi(t)$ and $\Psi(t)$ since their theoretical foundations, while our theoretical finding allows the modification because interpreting DDS as an optimization problem makes the identity preservation term a regularizer.** This modification not only reduces the editing time largely but also improves the quality. We carefully revised our manuscript to reflect the main differences compared to PDS. Please refer to lines 341-348, which have been highlighted in “red” for your convenience.
>
> ---
>
> > **[W2] Why DreamCatalyst is fast?**
>
> In common response 1, we clarify why DreamCatalyst can edit faster. Please refer to common response 1. We have carefully revised the manuscript accordingly to be clear as lines 341-348 and 376-377 as your thoughtful suggestions.
>
> In addition, **we have supplemented the quantitative and qualitative comparison results as suggested in Figures 17 and 18 to highlight DreamCatalyst’s convergence speed (please refer to the revised manuscript’s Appendix).** Figure 17 quantitatively demonstrates that DreamCatalyst converges significantly faster than the baseline methods. For this evaluation, we utilized CLIP Directional similarity as a metric to reflect the editing convergence behavior, since CLIP image similarity and Aesthetic score do not adequately capture the editing convergence behavior. Figure 18 presents qualitative results highlighting the editing convergence. These results indicate that DreamCatalyst achieves substantially faster convergence compared to PDS and IN2N. Thanks for your helpful feedback.

---

> ### Author Response · Authors · 2024-11-19
> **Response to Reviewer wbDz (Part 2/2)**
>
> > **[W3] Significance of FreeU**
>
> First, **the modified weighting of loss terms and FreeU are analogously important components in enhancing the editing quality of our framework.** As demonstrated in Tables 1 and 4, the performance improvements achieved by modifying the loss function and incorporating FreeU are comparable particularly in terms of CLIP-Directional Similarity and Aesthetic Score. Notably, $b=1.0$ in Table 4 indicates DreamCatalyst without FreeU (we revised Table 4 to clarify it), which already archives state-of-the-art results.
>
> For your convenience, we provide integrated quantitative comparisons from Tables 1, 4, and 6 as follows:
>
> | Method            | CLIP-Direc \($\uparrow$\) | CLIP-Img \($\uparrow$\) | Aesthetic \($\uparrow$\) |
> |---------------------|---------------------------|--------------------------|---------------------------|
> | DreamCatalyst                 | **0.180**                    | **0.746**                   | **5.688**                |
> | DreamCatalyst \(w/o FreeU\)     | *0.171*                    | *0.744*                   | *5.564*                    |
> | PDS \(w/ FreeU\)             | 0.162                | 0.668                   | 5.413                    |
> | PDS \(w/o FreeU\)             | 0.161                | 0.687                   | 5.437                    |
> | IN2N                 | 0.157                | 0.722                   | 5.399                    |
>
> Note that **Bold** indicates the best result and *Italic* is the second-best result.
>
> To further address your suggestion, **we conducted experiments applying FreeU to both PDS and DreamCatalyst** (please refer to Table 6 and Figure 19 in the revised manuscript’s Appendix). We set the FreeU hyperparameter $b=1.1$ for PDS, consistent with its configuration in DreamCatalyst. The findings are summarized below:
>
> - Applying FreeU to PDS: We observed a slight increase in the CLIP-Directional Similarity score (from 0.161 to 0.162), demonstrating the enhanced editability with utilizing FreeU. However, Aesthetic Score decreased (from 5.437 to 5.413). This is because **PDS underweights identity preservation at large timesteps as in Figure 2 (a), resulting in insufficient preservation of identity features.** Enhancing editability with FreeU further exacerbates this issue, causing the method to lose the original identity and generate unrealistic image results, such as over-editing and background distortions as shown in Figure 19.
>
> - Applying FreeU to DreamCatalyst: Integrating FreeU into our method significantly improved both the CLIP-Directional Similarity score (from 0.171 to 0.180) and the Aesthetic Score (from 5.564 to 5.688), demonstrating enhanced editability and visual quality. DreamCatalyst effectively balances editability and identity preservation by combining modified loss weighting and FreeU.
>
>
> These results indicate that **while FreeU can enhance editability metrics, its effectiveness depends on the underlying method's ability to preserve identity features and produce realistic images**. Therefore, combining our modified loss weighting with FreeU is essential for achieving superior results in DreamCatalyst.
>
> ---
>
> > **[Q1] Why is the approximated diffusion reverse process not compatible with PDS?**
>
> **The early stages of PDS with the approximated diffusion reverse process disrupt preserving the source features.** In PDS, identity preservation becomes insufficient at large timesteps due to the prioritization of editability. Consequently, SDS editing at large timesteps loses most of the source features because of strong noise perturbation and insufficient identity preservation as in Figure 2 (a). However, PDS using the approximated diffusion reverse process repeatedly samples large timesteps in the early stages of editing. As a result, the 3D model learns representations that lack source features, leading to a loss of the source identity. As shown in Figure 7, this produces results that resemble generation rather than editing, failing to preserve the source features. Therefore, PDS suffers from utilizing the approximated diffusion reverse process.
>
> ---
>
> > **[Q2] What is uniform sampling?**
>
> We provided an explanation of decreasing timestep sampling and its relationship to uniform sampling in common response 3. Please refer to the common response 3. As discussed in common response 3, each timestep $t$ is sampled multiple times, and we adopted a uniform sampling strategy to ensure equal sampling for every $t$.

---

> ### Author Response · Authors · 2024-11-23
> **Reminder**
>
> Dear Reviewer wbDz,
>
> Once again, thank you for taking the time to review our paper. We appreciate your efforts in helping us improve our work. We hope to inquire if you have any remaining concerns that we could address. Kindly let us know.
>
> Best regards,
>
> Authors

---

> ### Author Response · Authors · 2024-11-25
> **Sincerely looking forward to more discussion with you**
>
> Dear Reviewer wbDz,
>
> The discussion phase has only two days remaining, and we thus kindly request you to let us know if our response has addressed your concerns. If there are additional issues or questions, we would be happy to address them. Otherwise, we would greatly appreciate it if you could consider updating your score to reflect that the issues have been resolved.
>
> Best regards,
>
> Authors

---

> ### Author Response · Authors · 2024-11-26
> **Last day for revising the manuscript**
>
> Dear Reviewer wbDz,
>
> We sincerely appreciate your valuable suggestions. In accordance with your feedback, we have revised the manuscript and highlighted the changes in "red" as follows:
>
> - [W1] We clarified the difference between DreamCatalyst and PDS in the revised manuscript (lines 341-348).
> - [W2] We clarified the reason for the increased speed in the revised manuscript (lines 341-348 and 376-377). In addition, we have supplemented quantitative and qualitative comparisons in Figures 17 and 18.
> - [W3] We have included the ablation results of FreeU on DreamCatalyst and PDS in Table 6 and Figure 19.
>
> As today is the final day for revisions (with six days remaining in the discussion period), we wanted to inform you that there is limited time remaining before the revision period ends. If you have any further suggestions or concerns, please let us know at your earliest convenience. We are eager to discuss and address any additional feedback you may have.
>
> Best regards,
>
> Authors

---

> > ### Comment · Reviewer_wbDz · 2024-11-26
> >
> > I thank the authors for providing clarifications and further experiments to address the points raised in my review. I am convinced that the improvements gained through the ability to re-weight the two loss terms are significant. Additionally, the new explanation and figures showing why and to what extent this approach is faster than existing methods strengthens the submission. As a result of these improvements, I am raising my score to a 6.

---

> > > ### Author Response · Authors · 2024-11-27
> > >
> > > Could you kindly update the initial comments to reflect the revised score in the OpenReview system? We would sincerely appreciate it.

---

> ### Author Response · Authors · 2024-11-27
> **Response**
>
> We are glad that our rebuttal addressed your concerns, and we sincerely appreciate your decision to raise the score.
>
> Please don’t hesitate to let us know if you have any additional questions or feedback.

---

> ### Author Response · Authors · 2024-12-02
> **Updating the raised score**
>
> Dear Reviewer wbDz,
>
> As there are only 24 hours left in the review period, we kindly ask if you could update your review score in the OpenReview system to reflect your updated comments. We would greatly appreciate it if you could do so.
>
> Thank you very much for your time and consideration.
>
> Sincerely,
>
> Authors

---

> ### Author Response · Authors · 2024-12-04
>
> We would like to thank reviewer wbDz for the positive and constructive comments and raising the score throughout the review process. We are glad that we addressed the concerns.

---

### Author Response · Authors · 2024-11-19
**Common Responses**

Dear Reviewers and AC,

We sincerely appreciate the reviewers' thorough comments and constructive suggestions, which have greatly contributed to improving our work. As reviewers highlighted, we believe that DreamCatalyst achieves notably fast and high-quality results (wbDz, Hf3Z, p9uk) based on impressive theoretical reinterpretation (wbDz, Hf3Z) and provides extensive evaluations (wbDz, p9uk). In response to the feedback, we have carefully revised the manuscript as follows:

- We have clarified why DreamCatalyst achieves faster editing than the baseline methods and provided a comparison of convergence speeds.
- We have clarified the key differences between PDS and DreamCatalyst.
- We have conducted ablation studies on FreeU for both PDS and DreamCatalyst.
- We have supplemented the qualitative comparisons to teaser figures of PDS.

All revised content is marked using red-colored text for ease of identification.

Moreover, we have identified common questions raised by multiple reviewers and provided detailed responses to address each of these concerns comprehensively as follows.

---

> **Common response 1.  Why does DreamCatalyst save editing time? (reviewers wbDz and p9uk)**

(1) We emphasize that **the main factor in reducing the editing time is the weighting of the two primary loss terms.** Our timestep-dependent weighting strategy boosts editing speed for two reasons. First, the weighting condition enables using decreasing timestep sampling in the editing task. The decreasing timestep sampling allows fast score distillation because the sampling follows the diffusion denoising process. Second, our weighting avoids inefficient distillation. In small timesteps of PDS, the distillation of excessive identity preservation disturbs editing. However, our weighting makes efficient distillation by increasing the weight of editability at small timesteps, ensuring that the editing process remains uninterrupted.

(2) **Adopting FreeU instead of LoRA and Dreambooth also saves editing time.** While LoRA and Dreambooth demand extra computations for additional modules, FreeU does not require those.

Overall, our weighting of the two loss terms needs fewer optimization steps by considering the diffusion process and the role of timesteps, and FreeU requires less computation cost for each iteration.

---

> **Common response 2. The design of the special case in DreamCatalyst (reviewers Hf3Z and p9uk)**

Please note that our special case is based on the **two proposed conditions for the design choice of the formulation, which enables fast and high-quality editing with various function families.** We compared three cases, which follow the conditions, to verify the robustness of the design choice. As shown in Table 1, all three cases show state-of-the-art results compared to PDS and IN2N and show almost similar scores. This indicates our design rule is effective and robust.

However, we observed that inordinate editability in small timesteps rarely induces trivial color saturations on backgrounds (e.g., Figure 14’s “a skull face”). We hypothesize that the excessive editability during the final stages induces these color saturation artifacts. To prevent these color saturations, we designed $\Psi^{\*}(t)$ to drastically decrease editability in small timesteps. However, these artifacts from $\Psi^{\*}_2(t)$ and  $\Psi^{\*}_3(t)$ are rarely observed as the three cases in Table 1 show similar scores. Thus, **satisfying the two proposed conditions is the main key of DreamCatalyst.**

---

> **Common response 3. The difference between decreasing timestep sampling and DreamTime \[1\] (reviewers wbDz and p9uk)**

The primary distinction between decreasing timestep sampling and DreamTime lies in the **sampling rate to each timestep $t$.** SDS samples $t$ multiple times because the maximum timestep $T$ and the maximum number of iteration steps $N$ are not equal when $N > T$. DreamTime varies the sampling rate of $t$ by considering the timestep’s role. However, this sampling schedule makes it difficult to satisfy equation 15 (equation 12 in the revised manuscript) for each $t$ because some timesteps are optimized with fewer optimization steps. The equation 15 is defined as $\bar{x} = \arg\min_{x_0^{\text{tgt}}} \| \hat{x}_{0|t}^{\text{tgt}} - x_0^{\text{tgt}} \|^2$. Insufficient optimization steps for certain $t$ cause discrepancies in the diffusion sampling trajectory, as the optimization aligns SDS with this trajectory. To address this, we adopt a uniform sampling strategy, which samples $t$ uniformly as $t(i)= int(T(1 - i/N))$, ensuring that all timesteps are optimized equally throughout the process. By satisfying equation 15 for every $t$ with the uniform sampling, DreamCatalyst’s optimization aligns closely with the standard diffusion reverse process.

---

**Reference**

\[1\] Huang, Yukun, et al. "DreamTime: An Improved Optimization Strategy for Text-to-3D Content Creation." *arXiv preprint arXiv:2306.12422* (2023).

---

### Author Response · Authors · 2024-11-28
**Extended discussion period**

Dear Reviewers,

We sincerely appreciate the time and effort you have dedicated to reviewing our work. In response to your valuable feedback, we have thoroughly revised our submission to address the raised concerns and have included the suggested qualitative and quantitative analyses. As the revision phase is end and the extended discussion phase begins, we look forward to engaging in further discussion to clarify any remaining points. Thank you once again for your thoughtful comments and constructive suggestions.

Best regards,

Authors

---

### Author Response · Authors · 2024-12-03

Dear Reviewers and AC,

We sincerely appreciate your valuable feedback and constructive comments.

Reflecting on the reviews and discussions, we would like to highlight the key strengths of DreamCatalyst, which we believe address fundamental challenges in the field and demonstrate significant advancements:

---

> **Key Strengths of DreamCatalyst**

(1) **Theoretical Advancements and Generalized Framework**: As reviewers wbDz and Hf3Z highlighted our theoretical foundations, DreamCatalyst redefines the PDS framework by introducing a theoretically grounded objective function. This approach allows for dynamic reweighting of editing and identity preservation terms, **establishing PDS as a special case of DreamCatalyst**. This generalization not only enhances the theoretical foundation but also extends applicability beyond 3D editing, paving the way for regularization strategies across diverse editing tasks, as reviewer wbDz recognized the effectiveness of the reweighting. Furthermore, our theoretical foundation can also be extended to 4D editing, as mentioned in our response to [Q1]  from reviewer p9uk.

(2) **Substantial Gains in Quality and Practicality**: All reviewers (wbDz, Hf3Z, p9uk) emphasized that DreamCatalyst significantly enhances the efficiency and quality of 3D editing. In high-quality mode, it achieves state-of-the-art results across all quantitative metrics among the recent 3D editing methods. Additionally, the proposed fast mode accelerates editing speeds approximately **23 times faster than PDS**. These improvements streamline the superior performance and ensure practical usability for real-world applications.

(3) **First Application of FreeU to Editing Tasks**: DreamCatalyst integrates FreeU into the 3D editing pipeline **for the first time**, enhancing performance without compromising identity preservation as reviewer wbDz highlighted. Importantly, our proposed loss function alone achieves state-of-the-art results, and the addition of FreeU further amplifies these similar gains, demonstrating the synergy between our theoretical and architectural advancements.

---


Based on these strengths, we have carefully addressed the key concerns raised by the reviewers.

---

> **Distinct Contributions**

- Reviewer wbDZ noted that DreamCatalyst appears similar to PDS in the initial manuscript. As highlighted in **lines 341-348** of the revised manuscript (marked in red), in **response to [W1] from reviewer wbDz**, and in **common response 2**, we explicitly detailed how our theoretical advancements differentiate DreamCatalyst from PDS.
Notably, reviewer wbDz acknowledged the significance of these improvements and subsequently increased the score to a 6.

- Reviewer Hf3Z raised related concerns that DreamCatalyst might be perceived as a supplement to PDS or similar to other related methods. In **response to [W1] from reviewer Hf3Z**, we clarified that DreamCatalyst is distinct not only from PDS but also from works like Fantasia3D and ProlificDreamer.


---

> **Additional Experiments for Clarification**

- Reviewer wbDz suggested conducting experiments to highlight the editing speed and convergence behavior across iterations. As presented in **Figure 17 and 18**, we performed both quantitative and qualitative comparisons of convergence speed, demonstrating significantly faster convergence of DreamCatalyst compared to PDS.
- Reviewer Hf3Z proposed additional comparisons with results from the original PDS paper. In **Figure 16** and in **response to [W2] from Reviewer Hf3Z**, we included direct comparisons with teaser figures from the original PDS paper and project page, showing that DreamCatalyst consistently outperforms PDS.

---

> **Effect of FreeU**

- Reviewer wbDz suggested ablation studies to compare the effects of applying FreeU to both PDS and DreamCatalyst. In response to **[W3] from reviewer wbDz**, we updated **Table 1**, **Table 4**, added **Table 6**, and included **Figure 19**. These results demonstrate that DreamCatalyst without FreeU outperforms PDS with FreeU. As a result, reviewer wbDz raised the score due to the convincing results.

- Reviewer p9uk expressed concerns regarding the technical novelty of FreeU, given its broad applicability to diffusion models. In **response to [W1] from reviewer p9uk**, we clarified that DreamCatalyst is the first to unveil FreeU’s strength in editing tasks. As shown in **Table 6**, the performance gains from FreeU in DreamCatalyst are comparable to those achieved solely through our proposed loss function. Notably, PDS with FreeU fails to achieve such gains, underscoring the distinctiveness of DreamCatalyst.


We sincerely appreciate the reviewers’ recognition of the significance of our theoretical contributions and the promising advancements presented in our work. We are grateful for the positive recommendations from reviewer wbDz and reviewer p9uk and the constructive feedback from reviewer Hf3Z regarding our research.

Best regards,

Authors

---

### Meta-Review · Area_Chair_MJSt · 2024-12-22

**Metareview:**

The paper introduces DreamCatalyst, a method for fast and high-quality 3D editing. The key contribution of the paper is that it redefines the Posterior Distillation Sampling (PDS) framework. By introducing a theoretically grounded objective function, one can dynamic re-weight  the editing and identity preservation terms, making the original PDS a special case of DreamCatalyst. While there were originally concerns about the similarity between DreamCatalyst and PDS, as well as lack of details and comparisons, the authors put into a large amount of effort and addressed most of them. The reviews were on the fence pre-rebuttal. While one slightly negative reviewer stated that they will revise the score provided the authors address their issue, the reviewer did not respond during the discussion phase. The ACs looked into the response and can confirm that the authors indeed have answered most of the questions. The ACs hence assume the author will likely raise their score and the overall rating would be positive. The ACs urge the authors to incorporate the feedbacks from the reviewers into their final camera ready version.

**Additional Comments On Reviewer Discussion:**

The reviewers were mostly concerned about the difference to PDS, the role of FreeU, and the comparison to prior art. The authors provided comprehensive analyses in response.

---

### Decision · Program_Chairs · 2025-01-22

Accept (Poster)